# Comparative analysis of pattern-triggered and effector-triggered immunity gene expression in susceptible and tolerant cassava genotypes following begomovirus infection

**Bulelani L. Sizani¤, Keelan Krinsky, Oboikanyo A. Mokoka, Marie E. C. Rey**[ID]*

School of Molecular and Cell Biology, University of the Witwatersrand, Johannesburg, South Africa

¤ Current address: Biotechnology Platform, Agricultural Research Council, Onderstepoort, Pretoria, South Africa

* chrissie.rey@wits.ac.za

## Abstract

South African cassava mosaic virus (SACMV) is one of several bipartite begomoviruses that cause cassava mosaic disease (CMD) which reduces the production yield of the cassava (*Manihot esculenta* Crantz) crop in many tropical and subtropical regions. SACMV DNA-A and DNA-B encoded-proteins act as virulence factors that aid in inducing different disease severity depending on the host response. Recent evidence suggests a mutual potentiation of cell membrane receptor-associated pattern-triggered immunity (PTI) and nucleotide leucine-rich repeat (NLR) effector-associated immunity (ETI) in plant immune responses. This study aimed to compare expression of SACMV virulence factors, and PTI/ETI, in SACMV-infected susceptible T200 and tolerant TME3 cultivars. Expression of SACMV virulence factors differed between SACMV-infected T200 and TME3 plants at 12, 32 and 67 days post infection (dpi). Notably, at the early stage of infection (12 dpi), expression in TME3 of AV1 and AC2 virulence factors were 10-fold and 30-fold down-regulated, respectively, compared to susceptible T200. At systemic infection (32 dpi) AV1 expression was also significantly lower (4-fold) in TME3 compared to T200. Expression of AC2 (that targets host innate immunity), while significantly lower in both T200 and TME3 at 32 dpi compared to 12 dpi, was also significantly down-regulated (16-fold) in TME3 compared to T200. TME3 recovers around 67 dpi and virus load decreases by 33%, while in T200, symptoms and high SACMV replication persist. Identification and comparison of induced PTI and ETI associated genes upon SACMV-infection in susceptible T200 and tolerant/recovery TME3 cassava genotypes was achieved by whole transcriptome sequencing (RNA-seq) and by reverse transcriptase quantitative PCR (RT-qPCR). Analyses revealed reduced expression of PTI-associated signalling and response genes during SACMV systemic/symptomatic infection (32 dpi) in cassava genotypes. In addition, hydrogen peroxide ($H_2O_2$) production, a PTI indicator, was significantly reduced in the symptomatic viral infection stage at 32 dpi. Concurrently at 32 dpi, transcription of ETI signalling and response genes as well as SA biosynthesis and response genes, were upregulated during SACMV

**Data availability statement:** All relevant data are within the manuscript and its Supporting information files.

**Funding:** This project was awarded to the project leader MECR who funded the project which involved all authors (MECR, BLS, OM and KK). The project was within the framework of EraNet LEAP-AGRI (ID 18-LEAP-0004-03) (Europe Africa Research and Innovation (R&I) initiative related to Food and Nutrition Security and Sustainable Agriculture (FNSSA)). The specific research in this manuscript was funded by the French National Research Agency (ANR) (anr.fr/en/) through a grant agreement ANR-18-LEAP-0004-03 signed between ANR and WITS. Further funding for BLS was provided by the Carnegie Diversifying the Academy (programme, ID RKSBS21). The sponsors had no role in the study or manuscript.

**Competing interests:** The authors have declared that no competing interests exist.

systemic infection in TME3. These results indicate that SACMV targets PTI-associated genes during systemic infection at 32 dpi to subvert PTI-mediated antiviral immunity in cassava, which results in reduced induction of ROS production. Differential expression of specific NLR-associated genes also differed between susceptible and tolerant cultivars at 12, 32 and 67 dpi. SACMV virulence factors were shown to play a role in symptom severity in T200 and TME3.

## Introduction

Cassava is one of the staple crops in the world, providing energy intake for nearly a billion people [1] and supplying raw materials for various industrial uses worldwide [2]. In many regions of sub-Saharan Africa, cassava is considered a food security crop because of its relatively good performance in environmentally hostile conditions and flexible harvest period, as its roots can be left under the soil for a couple of years. Because of its high starch storage, cassava is becoming important as an industrial crop and livestock feed [2,3]. However, cassava yield is threatened by abiotic and biotic stresses, especially on the African continent. Cassava mosaic disease (CMD), cassava brown streak disease (CBSD), and cassava bacterial blight disease (CBBD), are the most prominent biotic diseases challenging cassava. Viral diseases have devastating effects as they are widespread [4], causing an average loss of up to 82% root yield in East Africa [4]. CMD is caused by several species of geminiviruses, including *South African cassava mosaic virus* (SACMV) [5], which are single-stranded bipartite circular DNA viruses composed of DNA-A and DNA-B components, belonging to the *Begomovirus* genus, family *Geminiviridae* [6]. As with other geminivirus-induced diseases, begomoviruses are transmitted mainly by the whitefly *Bemisia tabaci (*Gennadius) [6].

Begomoviral diversity emanates to a great extent from recombination [7], but also from the accumulation of point mutations [8]. While it has been argued that research into the ecology of begomovirus emergence has been neglected [9], there is evidence for co-adaptation of begomoviruses and whiteflies [10] that has led to speculation that changes in whitefly distribution and virus transmission may be linked to viral evolution. There is similarly evidence for viral evolution of begomoviruses in planta being significantly impacted by vegetative propagation [11]. In silico analysis suggests that interspecies recombination can result in significant diversity among geminiviruses and emergence of new geminivirus diseases and symptoms [12,13]. The recombination driven emergence of a more virulent begomovirus has already been reported for exchange of the C4 gene [13]. In cassava-infecting begomoviruses, recombination and variation may also arise in wild or non-cultivated hosts. Cassava mosaic begomoviruses comprise 11 species, with nine of them found in Africa shown to be recombinants [12]. *Indian cassava mosaic virus* (ICMV) and *Sri Lankan cassava mosaic virus* (SLCMV) are found in Asia with SLCMV being more virulent than ICMV [14]. It is interesting to note that SLCMV has characteristics of a monopartite begomovirus. For example, SLCMV can produce systemic infection by interacting with DNAβ component of *Ageratum yellow vein virus* (AYVV) to produce systemic infection in *Ageratum conyzoides*, and SLCMV DNA-A alone causes systemic infection resulting in upward leaf curling and stunting in *Nicotiana benthamiana* [14]. CMD leaf symptoms in general include mosaic, overall reduction in size, leaf curl or distortion, and yellowing and infected plant may also exhibit growth stunting [6]. To combat CMD, farmers grow disease resistant genotypes that are characterized to have CMD resistance loci.

Over the past decades, using quantitative trait loci (QTLs), two polygenic resistance loci (a recessive *CMD1* and a dominant *CMD3*) were identified in several Nigerian cassava landraces, and additionally, one dominant monogenic *CMD2* locus linked to CMD resistance was

identified [15,16]. *CMD2* is the most exploited locus due to its dominance and has been used extensively in breeding programs. Annotated *CMD2* locus-associated genes with different functions, including oxidative pathway-associated peroxidase enzymes, have been identified [15,16]. The class III peroxidase (POD) enzymes participate in plant development, hormone signalling, and stress responses. More recently, a study of a cassava genome revealed that there are 91 putative peroxidase genes with different expression profiles in storage root, stem, and leaves [17]. Plant class III peroxidases generate reactive oxygen species (ROS), and are reported to play a role in hydrogen peroxide ($H_2O_2$) induced immunity [18,19]. Kuon et al. [20] illustrated that resistance to geminiviruses in cassava is governed by haplotype inheritance, and Lim et al. [21] showed that genetic variation of DNA polymerase δ subunit 1 (*MePOLD1*) can explain resistance to geminiviruses in *CMD2*-type segregated populations. Resistance gene analogs have also been demonstrated to play a role in recovery in the CMD-tolerant TME3 landrace [22].

Bipartite geminivirus DNA-A and DNA-B encoded proteins act as virulence factors to assist the establishment of infection in host plants. Cassava mosaic geminivirus DNA-A genomes have six identified open reading frames (ORFs), encoding the capsid protein (*AV1*) and *AV2* on the virion strand, and *AC1, AC2, AC3,* and *AC4* ORFs being located on the complementary strand [6]. More recently, a geminivirus *AC5*-encoded protein was shown to interact with plant hormonal signalling and impact plant defense [23]. The DNA-B genome has two known ORFs, *BV1* encodes a nuclear shuttle protein (NSP), and *BC1* encodes a movement protein (MP) required for the cell-to-cell movement of the geminivirus [6]. The geminivirus capsid is built from 110 copies of the coat protein (CP), which also acts as an insect vector-specific determinant [10,24]. The AV2 proteins encoded by geminiviruses function as host post-transcriptional gene silencing (PTGS) inhibitors [25,26]. The AC2-encoded TrAP protein interferes with host transcriptional gene silencing (TGS) and PTGS resistance mechanisms [27]. The *AC1* encodes the replication initiation protein (Rep) protein, and the *AC3* ORF encodes the replication enhancer protein (REn) which are involved in geminiviral replication [28]. Geminiviral DNA-A and DNA-B genomes are reported to encode additional ORFs with virulence functions and specific subcellular localizations [29].

Similar to AC2, the AC4 protein is a suppressor of PTGS, TGS and cellular regulatory genes [30,31]. However, geminivirus C4/AC4 pathogenicity function is not only restricted to antiviral gene silencing during viral infection. The C4/AC4 proteins of geminiviruses accumulate at the intracellular plasma membrane and physically associate with the intracellular domain of the transmembrane receptor-like kinases (RLKs), and suppress the activation of mitogen-activated protein kinases (MAPK) cascades and their function in antiviral defence [32–34]. Remarkably, the AC4 interaction with receptor-like kinases hinders plant antiviral RNAi silencing. *Mungbean yellow mosaic virus* avirulence AC4 interacts with BARELY ANY MERISTEM1 (BAM1) family receptor kinases and suppresses PTGS by interfering with the cell-to-cell spread of siRNAs [35]. Geminiviral C4/AC4 proteins can also inhibit the (HIR1-mediated hypersensitive response (HR) often associated with resistance. For example, Mei et al. [36] showed that tomato leaf curl Yunnan virus (TLCYnV)-encoded C4 protein can induce the expression of *hypersensitive induced reaction 1 (HIR1)* and inhibit the self-interaction of the HIR1 protein to repress the HIR1-mediated HR through the physical interaction between C4 and HIR1. In addition, C4/AC4 proteins move from the plasma membrane (PM) to the chloroplast and subvert chloroplast-mediated host defence and salicylic acid-dependent antiviral responses [37,38]. Interestingly, the AC4 re-localization from PM to the chloroplast was also observed in cassava infected with *East African cassava mosaic virus*, with the *Coat Protein I (COPI)* pathway shown to be necessary for EACMV re-localization and induction of chloroplast-mediated infection [37].

PTGS involving small RNAs (siRNAs and miRNAs) has also been identified in host recovery to cassava begomoviruses [39–41]. To trigger resistance against pathogens, plants additionally have pattern recognition receptors (PRRs) and nucleotide-binding leucine-rich repeat containing receptors (NLRs). PRRs at the plasma membrane recognise pathogen-derived molecules to elicit pattern-triggered immunity (PTI), whilst intracellular localized NLRs (in the cytoplasm or nucleus) recognize pathogen effectors to elicit effector-triggered immunity (ETI). Originally, PTI was thought to function separately from ETI, however, most recent studies revealed crosstalk and cooperation between ETI and PTI [42].

The typical central nucleotide-binding site (NB-ARC) domain of the NLR proteins forms a globular shape and the horseshoe-like C-terminal entails leucine-rich repeats (LRR), whereas the N-terminal domains can either be a CC (coiled-coil) or a TIR (Toll-interleukin-1 receptor) domain. Generally, any domain of the NLR receptors can recognize viral effector molecules, directly or indirectly, leading to downstream activation of the signalling cascade. However, both TIR and CC domains contribute directly to pathogen invasion signalling and can induce cell death, autophagy or basal immunity. The TIR domain is a nicotinamide adenine dinucle-otide ($NAD^+$)-cleaving enzyme and has a 2'3'-nGMP synthetase activity involved in signalling and immunity [43–45]. CC domain signalling induces immunity through the formation of $Ca^{2+}$-permeable channels [46,47]. Quantitative trait loci (QTLs) for CMD resistance have been proposed [15,16], however to date, there are no published reports on NLR genes located in *CMD1*, *CMD2* and *CMD3* loci in cassava. Although there is significant understanding of downstream resistance (ETI) against plant viruses mediated by NLR proteins, only five NLR genes have been reported to confer resistance to geminiviruses to date [48,49].

The pathogen recognition receptors (PRRs) are cell surface-localized receptor-like kinases (LRR-RKs) or LRR receptor proteins (LRR-RPs) characterized by an extracellular LRR ligand-binding domain. The LRR domain of PRRs recognises molecular/pathogen-associated molecular patterns (MAMPs/PAMPs) and induces PTI responses, such as calcium flux, reactive oxygen species (ROS) burst, activation of mitogen-activated protein kinase (MAPKs) cascades, phytohormone homeostasis and regulation of defence genes [50–53]. MAMPs/PAMPs also induce callose deposition and cell death in leaves under biotic stress. Viral patterns recognized by PRRs in plants are usually double-stranded nucleic acids of viral origin [54] or encoded small proteins with virulence functions referred to as virus-associated molecular patterns (VAMPs) [55,56]. Nuclear shuttle protein-interacting kinase (NIK) is a transducer of plant defence signalling. NIK receptors belong to the plant defence group of the LRR receptor-like kinase (RLK) subfamily [57]. To date, NIK1 is the best characterized PRR and forms a complex with flagellin-sensing 2 (FLS2) and brassinosteroid insensitive (BRI1)-associated kinase 1 (BAK1) proteins to induce antiviral immunity [58]. Li et al. [58] showed that NIK1 antiviral signalling requires key downstream classical mitogen-activated protein kinases (MAPKs). MAP kinases are serine/threonine kinases that activate various effector proteins in the cytoplasm or nucleus, leading to ETI. FLS2/BAK1-mediated phosphorylation of NIK1 indicates that PRRs are crucial in plant immunity against geminiviruses. Phosphorylation of NIK1 is involved in the regulation of PTI key genes such as MAPKs, production of ROS, induction of PTI-associated marker genes and callose deposition in Arabidopsis [58].

PTI responses include homeostasis of plant defence hormone salicylic acid (SA) and upregulation of defence-related genes, in addition to high accumulation of ROS and MAPK activation. Notably, the pathogen-induced plant defence by SA is either positive or negative depending on the specific type of virus, pathogen, and host. SA confers resistance to plant viruses through SA signalling pathways centred on *NONEXPRESSER OF PR GENES 1 (NPR1)* and activation of *pathogenesis-related* (*PR*) genes associated with systemic acquired resis-tance (SAR) [59,60]. Interestingly, there are two distinct pathways for the biosynthesis of SA,

namely a phenylpropanoid pathway initiated by phenylalanine ammonia-lyase (PAL) in the cytosol and a chloroplastic pathogen-induced SA synthesis isochorismate synthase1 (ICS1) also known as SA INDUCTION DEFICIENT2 (SID2) [61,62]. PBS3 (avrPphB Susceptible 3) is another important enzyme important in SA biosynthesis and SA pathogen-induced plant response [61]. In addition, EPS1 (Enhanced Pseudomonas Susceptibility 1) was shown to play a role in pathogen-induced SA biosynthesis and together with PBS3, EPS1 acts in the cell cytosol downstream of PBS3 enzyme [61,62]. In Arabidopsis, ENHANCED DISEASE SUSCEPTI-BILITY5 (EDS5), a transmembrane protein (MATE transporter) extrudes isochorismate from the chloroplast to the cytoplasm and upon pathogen infection, *EDS5, ICS1, PBS3,* and *EPS1* gene expressions are induced [63]. PBS3 and EPS1 proteins are both necessary for ETI [61,63], and SA signalling is also required for PTI [61]. SA *de novo* biosynthesis is additionally positively regulated by WRKY transcription factors (WRKY TFs) through *ICS1* and *PBS3* [62].

Salicylic acid immune responses frequently result in downstream pathogen-induced hypersensitive response (HR) and systemic acquired resistance (SAR). HR is often, but not always, the initial plant immune response against pathogen invasion, by containing the spread of the pathogen at a primary infection site through induction of apoptosis in infected cells [64,65]. While this is shown in many annual/model crop studies, this has not been observed in CMD-infected perennial cassava. On the other hand, primary responses can induce systemic acquired resistance (SAR) in distal tissues providing protection against related and unrelated pathogens [62,63]. Nonexpresser of PR genes (NPRs) are involved in SAR, for example, NPR1-4 proteins mediate the expression of large-scale gene expression in response to SA homeostasis by interaction directly with many forms of immunity and cellular process regulatory pathways [61,63]. In Arabidopsis, NPR1 and NPR2 proteins are positive regulators of downstream SA response genes, whereas NPR3 and NPR4 act as negative regulators [63,65]. SA signal defence in plants, through NPRs, induce the expression of *PR (Pathogenesis-Related)* genes, which encode proteins with anti-microbial activities and are used as markers of the SA pathway [61].

An understanding of plant-virus interactions and dynamics is therefore critical for the development of strategies to manage the host response to virus infection under future, higher temperatures. Begomovirus pandemics in central East Africa have been linked to abundance of its insect vector, the whitefly, and they are correlated with historical climate change [66]. Synergistic interactions of the begomovirus-cassava system, together with other effects of climate change, are having a substantial negative impact on the food security of mostly smallholder farmers in Africa. Notable, an increase in temperature improving climate suitability for whiteflies has already been observed in central East Africa [66]. Such environmental pressures in turn can drive viral pathogenesis. Understanding the impact of elevated temperature on the molecular, cellular, physiological, and epidemiological dynamics of the cassava-infecting begomoviruses and their cassava host is therefore crucial to mitigate the effect of climate change on this pathosystem in Africa.

In plant-virus interactions, tolerance rather than resistance to pathogens have been reported, in particular in perennial crops, such as cassava [67]. In the TME3 landrace, tolerance to geminiviruses, including SACMV, has been reported [68], whereby symptoms and geminivirus replication persist, but are significantly reduced later in infection around 55–67 days post infection (dpi). In this study, we aimed to further explore the molecular basis for tolerance in the SACMV-TME3 pathosystem. Expression of SACMV virulence factors was compared between SACMV-infected susceptible T200 and tolerant TME3 genotypes at 12, 32 and 67 dpi. Herein, we also identified and compared expression of selected PTI and NLR-associated transcripts altered by SACMV in T200 and TME3, at the early stage of infection (12 days post-infection, dpi), when systemic virus replication is established (32 dpi), and at

67 dpi (when TME3 symptoms partially recover and virus replication declines, but severe symptoms and SACMV replication persist in T200). At 32 dpi when cassava plants were symptomatic, $H_2O_2$ and salicylic acid were measured in SACMV-infected T200 and TME3 compared to vector-only controls.

## Materials and methods

### Plant material, growth conditions and inoculation

Virus free cassava genotypes were maintained in tissue culture by nodal cutting propagation on Murashige and Skoog (MS) medium [69] containing 4.4 g.L$^{-1}$ MS salts, 20 g.L$^{-1}$ sucrose and solidified with 7.8 g.L$^{-1}$ plant agar, pH 5.8. For in vivo infections, cassava stem cuts were grown in MS medium containing 6.8 g.L$^{-1}$ of plant agar in a petri dish and grown for approximately 4 weeks at 25 °C under 16 h photoperiod (150 µEm$^{-2}$ sec$^{-1}$). After the stem cuttings developed sufficient root density, the plantlets were transferred to jiffy growth bags (Jiffies Products International) and placed on a deep tray covered with plastic film to maintain humidity and kept at 28 °C for approximately 4 weeks under the same photoperiod conditions before acclimatization. Acclimatized plantlets were allowed to grow until they developed at least two true leaves.

A total of 30 cassava genotypes were infected with SACMV to determine their phenotypic response to CMD (susceptible, tolerant or resistant). Plantlets were inoculated with 120 to 150 µl (OD = 600) of pBIN19-SACMV-DNA-A and pBIN19-SACMV-DNA-B infectious clones [5] transformed into *Agrobacterium tumefaciens* Agl1 as described by Allie et al. [67].

### Disease severity

Plants were visually assessed for cassava mosaic disease (CMD) symptoms on leaves over 3 months. Disease severity was performed using a scale of 1–5 described by Houngue et al. [70], where 1 = no leaves with symptoms characteristic of CMD; 2 = slight curl characteristic of CMD seen on leaves; 3 = CMD curling easily observable on leaves; 4 = CMD curling seen on many leaves; and 5 = very severe curling and leaf wilt.

### Viral load measurements

Viral replication levels of SACMV were measured at 12, 32 and 67 dpi by quantitative reverse transcription PCR (qPCR) [71] using core coat protein (CCP) primers (Table 1) targeting the CCP region on SACMV DNA-A component [5]. The selected time points (12 and 32 dpi) represent virus proliferation and symptomatic stages, respectively, and 67 dpi is the time point where TME3 is tolerant (recovery). Two to four fully expanded leaves below the shoot apex were collected as one biological replicate, and three biological replicates and three technical samples were used and immediately frozen in liquid and stored at − 80 °C until further use.

### Measurement of hydrogen peroxide production

Hydrogen peroxide ($H_2O_2$) production was measured colormetrically according to Xu et al. [72]. Leaves below the shoot apex from SACMV infected and mock (empty pBIN19 clones) in T200 and TME3 plants were collected and kept at − 80 °C and used within seven days. Two to four fully expanded leaves were collected as one biological replicate, and three biological replicates and three technical samples were used. Homogenized leaf tissue was mixed immediately with 10 mM Tris-HCl buffer (pH 7.3) in a ratio of 1: 50 in an ice bath. The extract was centrifuged at 15,000 rpm for 5 min, and 500 µl of the supernatant was added to 500 µl of 10 mM potassium phosphate buffer (pH 7.0) and 1 ml of 1 M potassium iodide. The absorbance of the

supernatant was read at 390 nm. The levels (mol.mg$^{-1}$ FW) of $H_2O_2$ were ascertained from a standard curve.

## Transcriptome analysis and quantitative reverse transcription PCR

The transcriptome of CMD-susceptible cassava T200 and CMD-tolerant TME3 landraces at 12, 32 and 67 dpi infected with SACMV compared to mock-inoculated was previously reported [67]. Differentially expressed (DE) transcripts originally identified from alignment with the reference cassava AM560-2 genome v4.1 were remapped to the updated cassava *AM560-2* v8.1 genome (https://phytozome-next.jgi.doe.gov/info/Mesculenta_v8_1, 2024). DE genes associated with PTI, ETI and SA upon SACMV-infection were subsequently identified.

Reverse transcription quantitative PCR (RT-qPCR) for selected PTI, ETI, SA and SACMV virulence genes (Table 1) was performed as follows: the leaf samples from SACMV- and mock-infected plants were collected at 12, 32 and 67 dpi and kept at −80 °C until further use. Two to four fully expanded leaves below the shoot apex were collected as one biological replicate, and three biological replicates, each comprising three technical samples were used. Total RNAs were extracted by using the CTAB (2% CTAB, 2% PVP-40, 20 mM Tris–HCl, pH 8.0, 1.4 M NaCl, 20 mM ethylenediaminetetraacetic acid) extraction protocol according to Behnam et al. [73]. Two hundred ng (200 ng) of total RNA was treated with DNase I (New England Biolabs, Ipswich, USA), and cDNA was reversely transcribed with M-MuLV reverse transcriptase (Thermo Fisher Scientific, Waltham, USA). RT-qPCR analysis was performed with CFX Manager (Bio-Rad, Hecrcules, USA) using 2X Luna Universal qPCR Master Mix (New England Biolabs, Ipswich, USA). *UBQ10* (*Ubiquitin 10*) and *GTPb* (*GTP binding*) genes served as the endogenous control as previously described by Moreno et al. [74].

**Table 1. Primers used for SACMV DNA-A and DNA-B virulence proteins and PacBio SMRT sequence of NLR and LRR-PK/PL genes.**

| Primers used in SMRT sequencing | | | | |
|---|---|---|---|---|
| | **Gene Name** | **F-primer** | **R-primer** | **Length (bp)** |
| SACMV DNA-A & DNA-B virulence genes | *AV1 (CP)* | CAGGCTTGGTGGGAGATT | AGCGTAGCATACACTGGATTAG | 75 |
| | *AC2* | GTCGTGGTTGGTGATGTCGA | TCGACTGCATCAACCATGGA | 120 |
| | *AC4* | CTTGGCGTAAGCGTCATTGG | CCAATGGGGCGAGTTTCAGA | 76 |
| | *BC1* | ATAACCGACCCAGTTGCGTT | CAGGCTACCATGCGACTCAA | 98 |
| | *BV1* | CTTTCGTTCCAGAGGGGGTC | AACCTAGGCATGCCGTACAC | 85 |
| NLR and LRR-PK/PL genes | **Gene Name** | **R gene type** **F-primer** | **R-primer** | **Length (bp)** |
| | *Manes.07G004200* | LRR CTTCAGACACACCTCCTGCA | CACATGCATGCAACTGGCTT | 1,347 |
| | *Manes.11G060000* | LRR AGCACTCGCAAATAGATATATAAAAG | ACCAGAAGCAAAATAAGGGCA | 2,379 |
| | *Manes.18G066400* | LRR GTGTTACATGCGCGTACGTC | CGCATGCCATTCCACTAGGA | 3,591 |
| | *Manes.10G134000* | NB-LRR CTGCCAAAGCTTGCTTCCAA | CCTCATTGTTGGAGTGCTGC | 4,551 |
| | *Manes.12G091600* | NB-LRR AGCTCTTGTGTATGGGTGCC | GGGTTCCCAGTCTCAACAGG | 1,840 |
| | *Manes.05G169600* | CC-NB TCGGCATATCTTCACACGCC | TTTCTACCAGTATCAACAGAG-TATCA | 3,112 |
| | *Manes.S063200* | CC-NB-LRR GCATCAAAGGAACAAGCGCA | ATCCTGGGTTGGACTTCTGC | 3,193 |
| | *Manes.10G127900* | TIR-NB-LRR GCTCAGTTGGAGTTACACAAGC | ACTCAACAATCTGTACCATGGA | 3,486 |
| | Adaptors | GCAGTCGAACATGTAGCTGACTCAG-GTCAC* | TGGATCACTTGTGCAAGCATCA-CATCGTAG* | |

*Amino acid added at 5' end to block unintended amplification.

## Library construction for single molecule real-time (SMRT) sequencing

The NLR libraries were generated from 24 cassava genotypes genomic DNA by PCR using Phusion High-Fidelity DNA Polymerase (Thermo Fisher Scientific, Waltham, USA) using primers listed in Table 1. Two-step PCR was conducted for library construction. The first PCR was performed on genomic DNA to amplify selected NLR genes using PCR primers designed to include a common sequence for annealing in the second PCR along with the gene-specific region. The second set of primers (Table 1) was designed to synthesize PCR products with PacBio Sequel II System (Pacific Bio-science Inc., Menlo Park, USA) adaptors and barcodes for SMRT sequencing to identify each sample. All amplicons were pooled in a single cell for HiFi reads sequencing using equal quantities of each molecule. The required volume of each amplicon was calculated according to DNA concentration. The HiFi reads were assembled with HiCanu [75] and available on NCBI under BioProject ID: PRJNA1190476.

## Structural analysis and molecular modelling

Protein NLR domains were identified based on the Plant Resistance Gene database (PRGdb; http://prgdb.org/prgdb4/, 2023). Protein secondary structure-based alignment was done based on *NRG1* for helper NLRs (PBD ID 7l7w; [47] and or *ZAR1* for CNL (PDB ID 6j5w; [76]. For TIR domains, the consensus of RUN (PDB ID 7rx1; [43], RPV1 (PDB ID 5ku7; [77], and RPP1 (PDB ID 5teb; [78] was built to increase the prediction reliability. Alignments were created using PROMALS3D (http://prodata.swmed.edu/promals3d/promals3d.php, 2023) and homology 3D models were built using Modeller v10 (https://salilab.org/modeller/, 2023). Visual inspection and protein structure image were performed with UCSF ChimeraX (https://www.rbvi.ucsf.edu/chimerax/, 2024). Models were further subjected to structural validation using MolProbity (http://molprobity.biochem.duke.edu/, 2024).

## Statistical analysis

Statistical analysis of viral load and quantitative reverse transcription PCR were conducted by *t*-test in Microsoft Excel. Accumulation of ROS ($H_2O_2$) in infected leaves was conducted by a two-way ANOVA followed by LSD tests (P < 0.05) using RStudio v4 (https://www.r-project.org/).

# Results

## Symptoms of cassava mosaic disease caused by South African cassava mosaic virus

Four cassava (*Manihot esculenta* Crantz) genotypes sourced from western and southern Africa were screened for CMD response to SACMV, a bipartite genome begomovirus (Fig 1a). Inoculation of these selected cassava genotypes with SACMV displayed susceptibility, resistance or tolerance responses (Fig 1b and S1 Table) indicating that cassava genotypes exhibit differential responses depending on their genetic source. In all genotypes, symptoms were non-detectable at 12 dpi, however at 32 dpi T200 and TME3 became highly symptomatic (symptom index = 3, S1 Table). Susceptible Ukulinga 9 had a delayed symptom response which manifested approximately around 50 dpi and persisted at 67 dpi (Fig 1b and S1 Table). TMS98/0505 displays no symptoms at any measured time point post-infection.

## Virus replication

To determine whether the delayed susceptibility response in Ukulinga 9 is due to a slower accumulation of SACMV over time and whether the recovery phenomenon in TME3 is due

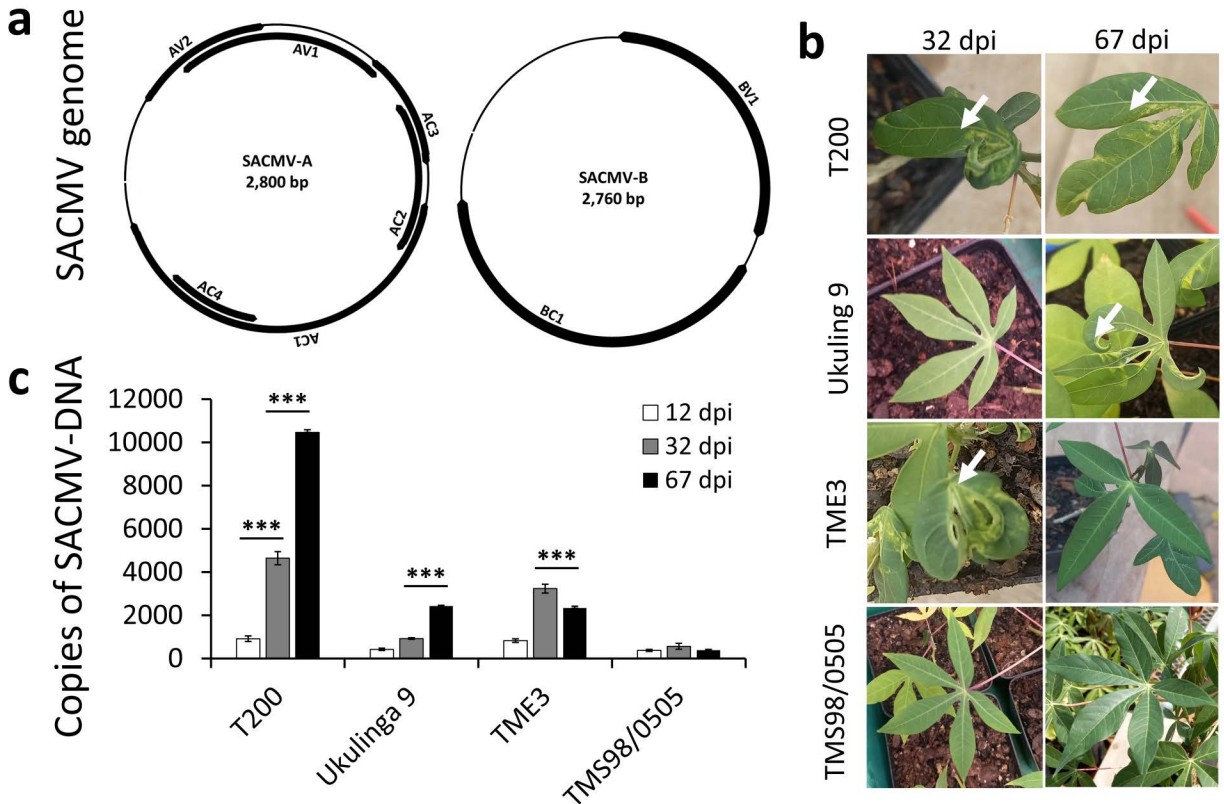

**Fig 1. South African cassava mosaic virus (SACMV) genome structure and the effect of SACMV on cassava leaf symptoms. (a)** A bipartite ssDNA-A and ssDNA-B genomic structure of SACMV. AC1: replication-associated protein (Rep); AC2: transcriptional activator protein (TrAP); AC3: replication enhancer protein (REn); AC4: silencing suppressor protein); AV1: coat protein (CP); AV2: pre-coat protein; BC1: movement protein (MP) and BV1: nuclear shuttle protein (NSP). **(b)** Leaf symptoms induced by SACMV manifested by leaf curl and yellow mosaic (arrows). In susceptible T200 symptoms persist at 32 and 67 dpi, while in Ukulinga 9 symptoms appear after 32 dpi (delayed susceptible phenotype). TME3 symptoms manifested at 32 dpi (tolerant, recovery phenotype) but the plant recovered at 67 dpi. TMS98/0505 exhibits no symptoms at any given time point post-infection (resistant genotype). **(c)** Viral load in T200, Ukulinga 9, TME3 and TMS98/0505 at 12 (pre-symptomatic stage), 32 (symptomatic stage) and 67 (recovery stage) dpi determined as DNA copies of CP using qPCR. Data represent the mean of three independent biological replicates. Error bars represent SD. An asterisk indicates a statistically significant difference according to unpaired Student's t-test (two-tailed), *$p \leq 0.05$. GTPb was used as a housekeeping gene.

to its ability to suppress or restrict viral replication, copies of the CP gene (*AV1*) at 12, 32 and 67 dpi were determined by qPCR. Susceptibility coincided with an increase in SACMV copies in T200, TME3 and Ukulunga 9 over time, whereas there was no significant accumulation of SACMV at any time points post-infection in the resistant TMS98/0505 genotype (Fig 1c). Notably, at 67 dpi, TME3 partially recovers showing an attenuation in symptom severity, and exhibits a concomitant marginal reduction in viral load accumulation compared to 32 dpi (Fig 1c).

## Total number of RNA transcripts differentially expressed in T200 and TME3 infected with SACMV at 12, 32 and 67 dpi

Expression profiling of NLR genes was compared between susceptible T200 and tolerant TME3 at 12, 32 and 67 dpi. A total of 17,235 RNA transcripts were DE in T200 and TME3 at all three time points. In T200, a total of 14,029 (81%) transcripts were DE, whereas notably only 3,206 (18%) of transcripts were DE in TME3 (Fig 2a). The total number of DE RNA

transcripts in T200 increased from 3,415 at 12 dpi to 5,242 at 32 dpi (65%) and remained constant thereafter (67 dpi). In contrast, in TME3, the total number of DE RNA transcripts progressively decreased over time (Fig 2a). Similarly, the total number of DE NLR genes in T200 was significantly higher at all time points compared with TME3 (Figs 2b–e). In TME3, the number of DE NLR transcripts were at their highest at early infection (12 dpi) and decreased to only five and four at 32 and 67 dpi, respectively (Figs 2b–e). Notably, almost half of NLR genes (53%) expressed in tolerant TME3 were also expressed in susceptible T200 (Fig 2f) indicating that SACMV (geminiviruses) infection has induced a comparable response in the susceptible and tolerant cassava genotype.

## Expression levels of SACMV DNA-A and DNA-B virulence factors in TME3 and T200 at 12, 32 and 67dpi

Geminiviruses encode proteins with specific subcellular localizations and virulence functions [29]. Gene expression of SACMV genes, whose proteins can act as pathogenicity determinants (virulence factors) (Table 2), was measured in susceptible T200 and tolerant TME3 plants infected with SACMV at (i) the time point (12 dpi) when plants show no symptoms but SACMV is just starting to replicate and move; at (ii) 32 dpi when plants are symptomatic and SACMV replication is high, and (iii) at 67 dpi when symptoms/virus replication persist in T200 but have started to decline/recover in TME3. The expression of SACMV-encoded virulence factors *AV1, AC2, AC4, BC1* and *BV1* transcripts varied in and between both genotypes. Notably, geminivirus virulence genes *AV1* and *AC2*, reported to induce systemic infection by targeting host gene silencing mechanisms (RNAi), were highly expressed early at 12 dpi in the susceptible T200 cultivar compared to the tolerant TME3 cultivar. Expression of *AC4* was significantly higher in TME3 than in T200 at 32 dpi. *BV1* and *BC1*, that are associated with subverting PTI-mediated immunity, were highly expressed in TME3 at the symptomatic stage (32 dpi), but at the symptom recovery/decrease in SACMV replication stage (67 dpi), their expression significantly decreased compared to T200 (Table 2). The transcript levels of the nuclear shuttle protein (NSP), a pathogen-determinant viral protein encoded by the *BV1* (Fig 1a), were significantly lower in TME3 compared to T200 at 67 dpi (Fig 1c).

## Comparison of pathogen-triggered immunity, effector-triggered immunity, and salicylic acid-dependent immune responsive genes in TME3 symptomatic and recovered leaves

To determine whether PTI mediates antiviral immunity to SACMV infection, gene expression of key PTI genes at 12, 32 and 67 dpi was determined (Figs 3a–c). During the early stage of infection, MAPK key and indicator genes were downregulated in both susceptible T200 and tolerant TME3 and remained predominantly downregulated in T200 throughout the duration of this study. In TME3, the MAPK transcripts were upregulated at 32 dpi (Fig 3b). Transcripts of NIK key and response genes were upregulated in TME3 only at 12 dpi, while in T200 they were upregulated at 32 and 67 dpi. ETI key and response genes were downregulated in T200 at all the time points, but only at 12 dpi in TME3, and no differential regulation at 32 and 67 dpi was observed (Figs 3a–c). Since SA is involved in both PTI and ETI plant immune response and is a point of convergence of PTI and ETI [61], transcripts of SA were analysed. SA biosynthesis and indicator genes were downregulated in T200 at 12 dpi and remained downregulated at 32 and 67 dpi (Figs 3a–c). However, in TME3 few SA transcripts were DE only at 12 dpi, and no change of expression thereafter (Figs 3a–c). Quantitative gene expression (RT-qPCR) analysis indicated that PTI key and indicator genes, particularly NIK-pathway genes, were

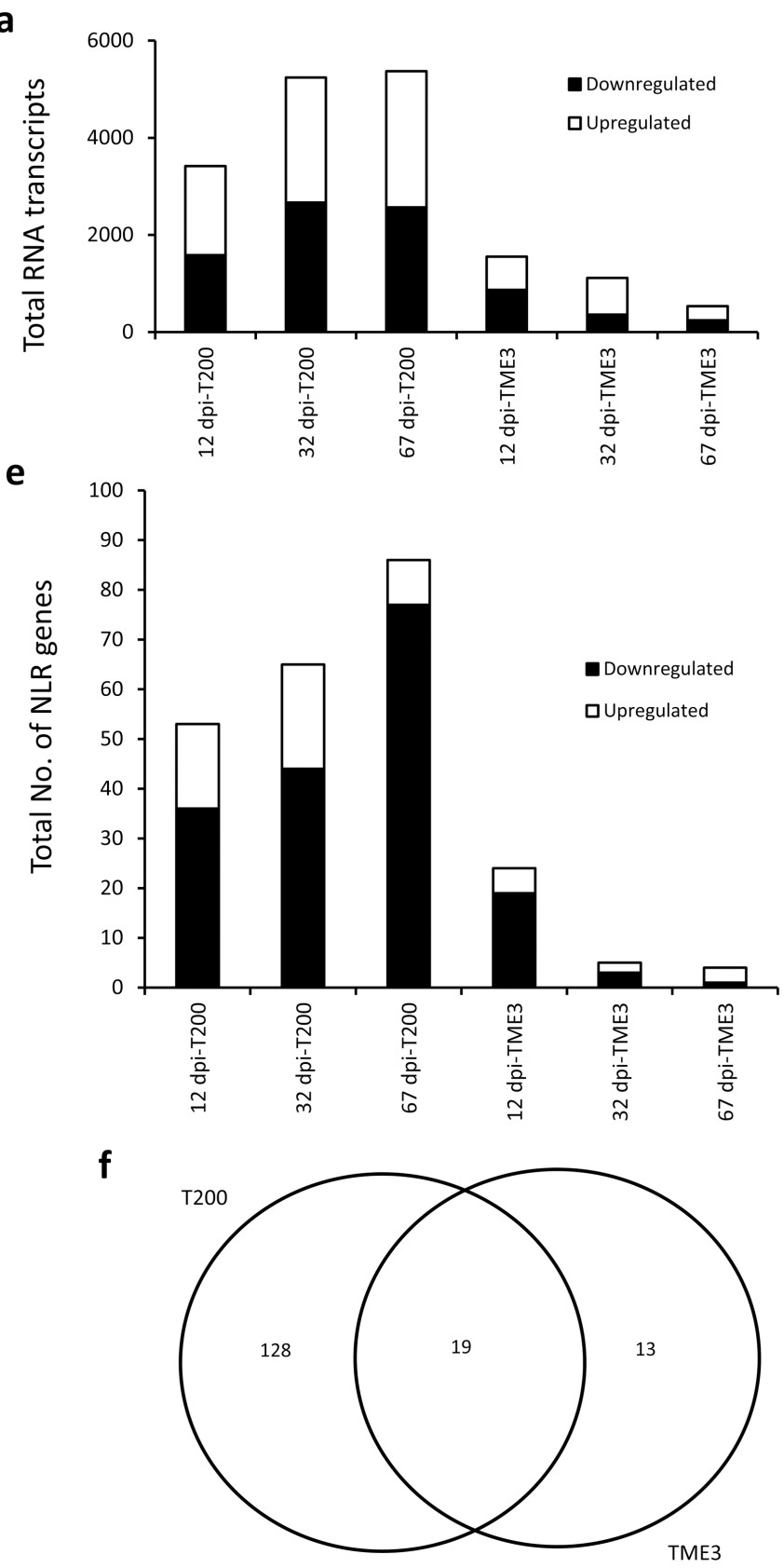

**b**

| T200 12 dpi | | |
|---|---|---|
| **Gene ID** | **Class** | **log FC** |
| Manes.11G053100 | NL | 4,2 |
| Manes.10G028400 | NL | 7,1 |
| Manes.07G062324 | NL | 6,6 |
| Manes.02G201301 | NL | 4,5 |
| Manes.02G209475 | NL | 4,3 |
| Manes.03G036300 | NL | 5,9 |
| Manes.02G215600 | NL | 7,7 |
| Manes.02G215120 | NL | 10,2 |
| Manes.10G127900 | NL | 15,9 |
| Manes.07G009936 | RNL | 4,8 |
| Manes.07G008990 | RNL | 4,2 |
| Manes.02G097400 | RNL | 8,2 |
| Manes.07G060833 | CNL | 4,9 |
| Manes.11G122816 | CNL | 17,5 |
| Manes.11G122832 | CNL | 19,1 |
| Manes.11G122860 | CNL | 15,5 |
| Manes.11G122880 | CNL | 21,0 |
| Manes.10G028600 | CNL | 5,4 |
| Manes.10G139900 | CNL | 13,7 |
| Manes.07G043891 | CNL | 3,9 |
| Manes.07G121300 | CNL | 4,3 |
| Manes.07G009201 | CNL | 4,2 |
| Manes.02G215425 | CNL | 28,1 |
| Manes.02G215445 | CNL | 23,3 |
| Manes.02G215700 | CNL | 16,5 |
| Manes.09G025800 | CNL | 5,1 |
| Manes.02G214775 | CNL | 12,4 |
| Manes.09G019900 | CNL | 4,0 |
| Manes.02G215400 | CNL | 4,1 |
| Manes.10G025400 | CNL | 4,0 |
| Manes.07G107800 | CNL | 5,9 |
| Manes.09G019801 | CNL | 7,2 |
| Manes.10G091500 | CNL | 4,3 |
| Manes.02G215420 | CNL | 5,5 |
| Manes.02G215615 | CNL | 5,2 |
| Manes.04G165600 | CNL | 4,7 |
| Manes.02G220660 | CNL | 14,4 |
| Manes.02G215405 | CNL | 13,2 |
| Manes.02G215000 | CNL | 11,8 |
| Manes.10G144000 | CNL | 10,7 |
| Manes.02G215430 | CNL | 8,7 |
| Manes.14G148375 | TNL | 4,8 |
| Manes.10G121812 | TNL | 14,1 |
| Manes.18G125400 | TNL | 11,3 |
| Manes.12G149700 | TNL | 4,6 |
| Manes.02G202166 | TNL | 4,1 |
| Manes.10G127266 | TNL | 13,3 |
| Manes.10G125200 | TNL | 12,5 |
| Manes.11G061000 | TNL | 10,1 |
| Manes.10G128001 | TNL | 4,8 |
| Manes.10G127950 | TNL | 19,3 |
| Manes.10G126600 | TNL | 20,7 |
| Manes.10G126900 | TNL | 27,5 |

| TME3 12 dpi | | |
|---|---|---|
| **Gene ID** | **Class** | **log FC** |
| Manes.18G136808 | N | 5,7 |
| Manes.11G129900 | NL | 5,7 |
| Manes.15G123000 | NL | 9,6 |
| Manes.10G074374 | NL | 3,9 |
| Manes.07G044166 | RNL | 4,1 |
| Manes.05G169600 | RNL | 13,3 |
| Manes.02G097400 | RNL | 9,0 |
| Manes.09G019801 | CNL | 4,9 |
| Manes.18G117100 | CNL | 4,4 |
| Manes.07G062340 | CNL | 4,4 |
| Manes.10G133653 | CNL | 12,4 |
| Manes.10G091700 | CNL | 5,8 |
| Manes.10G128000 | CNL | 8,0 |
| Manes.09G025500 | CNL | 9,1 |
| Manes.10G133710 | CNL | 5,5 |
| Manes.10G127266 | TNL | 5,2 |
| Manes.10G126900 | TNL | 4,5 |
| Manes.10G125200 | TNL | 4,0 |
| Manes.10G126732 | TNL | 7,0 |
| Manes.S095306 | TNL | 4,2 |
| Manes.14G148375 | TNL | 6,8 |
| Manes.09G158500 | TNL | 4,7 |
| Manes.02G199800 | TNL | 7,4 |
| Manes.06G161500 | TNL | 13,7 |

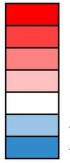

**c**

| T200 32 dpi | | |
|---|---|---|
| **Gene ID** | **Class** | **log FC** |
| Manes.02G201200 | NL | 10,9 |
| Manes.18G112400 | NL | 4,0 |
| Manes.08G161520 | NL | 6,8 |
| Manes.07G060716 | NL | 3,8 |
| Manes.09G084300 | NL | 5,1 |
| Manes.05G063200 | NL | 29,6 |
| Manes.05G116000 | NL | 22,5 |
| Manes.02G215600 | NL | 20,8 |
| Manes.02G202959 | NL | 4,9 |
| Manes.05G097902 | RNL | 5,6 |
| Manes.07G044350 | RNL | 8,8 |
| Manes.13G128600 | RNL | 23,4 |
| Manes.05G169600 | RNL | 80,0 |
| Manes.11G122816 | CNL | 16,5 |
| Manes.11G122880 | CNL | 14,1 |
| Manes.10G028600 | CNL | 12,1 |
| Manes.11G122832 | CNL | 11,9 |
| Manes.07G009201 | CNL | 8,0 |
| Manes.11G122800 | CNL | 7,3 |
| Manes.07G062324 | CNL | 7,1 |
| Manes.07G008700 | CNL | 6,0 |
| Manes.07G062404 | CNL | 5,9 |
| Manes.10G128000 | CNL | 4,6 |
| Manes.10G028400 | CNL | 4,2 |
| Manes.07G062348 | CNL | 5,2 |
| Manes.10G022500 | CNL | 5,2 |
| Manes.02G220660 | CNL | 55,6 |
| Manes.02G215445 | CNL | 53,8 |
| Manes.02G215425 | CNL | 62,3 |
| Manes.02G215700 | CNL | 31,3 |
| Manes.02G215615 | CNL | 27,3 |
| Manes.02G215420 | CNL | 16,3 |
| Manes.02G215405 | CNL | 45,5 |
| Manes.02G215400 | CNL | 25,9 |
| Manes.02G215120 | CNL | 44,8 |
| Manes.02G215000 | CNL | 31,1 |
| Manes.02G215015 | CNL | 14,8 |
| Manes.02G214775 | CNL | 8,6 |
| Manes.10G135900 | CNL | 13,8 |
| Manes.09G019900 | CNL | 13,5 |
| Manes.02G107000 | CNL | 13,4 |
| Manes.11G123200 | CNL | 13,2 |
| Manes.02G215010 | CNL | 10,2 |
| Manes.09G019701 | CNL | 9,9 |
| Manes.09G019801 | CNL | 9,2 |
| Manes.14G169600 | CNL | 5,8 |
| Manes.10G025400 | CNL | 5,3 |
| Manes.06G013900 | CNL | 5,3 |
| Manes.07G043800 | CNL | 4,3 |
| Manes.09G025130 | CNL | 3,7 |
| Manes.18G112450 | TNL | 9,4 |
| Manes.10G127500 | TNL | 7,1 |
| Manes.S095306 | TNL | 7,4 |
| Manes.06G012900 | TNL | 13,6 |
| Manes.14G148375 | TNL | 9,4 |
| Manes.06G161500 | TNL | 9,2 |
| Manes.02G199800 | TNL | 17,7 |
| Manes.10G128001 | TNL | 76,6 |
| Manes.10G127975 | TNL | 35,7 |
| Manes.10G126732 | TNL | 24,9 |
| Manes.10G121812 | TNL | 150,0 |
| Manes.10G126600 | TNL | 335,4 |
| Manes.10G126900 | TNL | 233,7 |
| Manes.10G127950 | TNL | 172,5 |
| Manes.18G112080 | TNL | 20,5 |

| TME3 32 dpi | | |
|---|---|---|
| **Gene ID** | **Class** | **log FC** |
| Manes.03G060801 | NL | 7,0 |
| Manes.05G169600 | RNL | 10,0 |
| Manes.18G105800 | CNL | 6,8 |
| Manes.10G133683 | CNL | 7,9 |
| Manes.07G063001 | CNL | 4,6 |

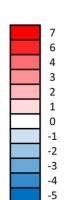

**d**

| T200 67 dpi | | |
|---|---|---|
| Gene ID | Class | log FC |
| Manes.11G053100 | NL | 7,2 |
| Manes.08G161520 | NL | 10,8 |
| Manes.05G063200 | NL | 16,7 |
| Manes.05G116000 | NL | 16,9 |
| Manes.10G127900 | NL | 169,1 |
| Manes.02G215600 | NL | 57,0 |
| Manes.10G141651 | NL | 4,1 |
| Manes.07G060700 | NL | 11,5 |
| Manes.03G114816 | NL | 10,7 |
| Manes.02G221201 | NL | 5,9 |
| Manes.10G114062 | NL | 10,0 |
| Manes.08G163700 | NL | 5,4 |
| Manes.18G144600 | NL | 5,1 |
| Manes.03G063225 | NL | 5,8 |
| Manes.04G055000 | NL | 5,7 |
| Manes.07G044166 | RNL | 4,0 |
| Manes.13G036800 | RNL | 4,3 |
| Manes.13G128600 | RNL | 19,7 |
| Manes.05G169600 | RNL | 66,4 |
| Manes.04G103400 | RNL | 16,8 |
| Manes.10G139900 | CNL | 10,5 |
| Manes.11G122816 | CNL | 25,1 |
| Manes.11G122860 | CNL | 15,4 |
| Manes.11G122832 | CNL | 11,5 |
| Manes.07G011300 | CNL | 3,8 |
| Manes.07G062324 | CNL | 4,2 |
| Manes.07G062404 | CNL | 3,9 |
| Manes.09G025560 | CNL | 10,0 |
| Manes.09G034700 | CNL | 6,1 |
| Manes.09G025800 | CNL | 4,7 |
| Manes.10G141800 | CNL | 6,5 |
| Manes.06G013900 | CNL | 14,3 |
| Manes.10G135900 | CNL | 13,1 |
| Manes.10G144000 | CNL | 7,8 |
| Manes.09G025130 | CNL | 7,1 |
| Manes.14G165300 | CNL | 6,9 |
| Manes.11G123200 | CNL | 6,6 |
| Manes.10G133710 | CNL | 6,6 |
| Manes.07G062348 | CNL | 6,5 |
| Manes.10G137900 | CNL | 13,0 |
| Manes.09G025388 | CNL | 5,9 |
| Manes.07G060300 | CNL | 12,5 |
| Manes.09G025044 | CNL | 10,6 |
| Manes.11G129700 | CNL | 5,1 |
| Manes.07G062340 | CNL | 4,6 |
| Manes.07G107800 | CNL | 4,4 |
| Manes.02G107000 | CNL | 4,2 |
| Manes.07G045400 | CNL | 4,0 |
| Manes.09G022500 | CNL | 3,9 |
| Manes.02G215615 | CNL | 73,6 |
| Manes.02G215700 | CNL | 58,2 |
| Manes.02G220660 | CNL | 75,4 |
| Manes.14G169600 | CNL | 42,8 |
| Manes.02G215015 | CNL | 19,4 |
| Manes.02G215010 | CNL | 38,9 |
| Manes.02G214775 | CNL | 20,0 |
| Manes.09G019900 | CNL | 23,7 |
| Manes.09G019801 | CNL | 18,2 |
| Manes.09G019701 | CNL | 38,0 |
| Manes.07G062308 | CNL | 16,6 |
| Manes.07G061500 | CNL | 34,1 |
| Manes.02G215000 | CNL | 92,6 |
| Manes.02G215120 | CNL | 48,0 |
| Manes.02G215400 | CNL | 52,9 |
| Manes.02G215405 | CNL | 66,3 |
| Manes.02G215420 | CNL | 40,2 |
| Manes.02G215425 | CNL | 67,9 |
| Manes.02G215445 | CNL | 83,0 |
| Manes.18G108400 | TNL | 6,0 |
| Manes.10G126732 | TNL | 4,1 |
| Manes.S095306 | TNL | 3,8 |
| Manes.06G012900 | TNL | 12,4 |
| Manes.18G114900 | TNL | 12,3 |
| Manes.02G202166 | TNL | 6,6 |
| Manes.18G115500 | TNL | 4,4 |
| Manes.11G061000 | TNL | 13,8 |
| Manes.09G158500 | TNL | 9,5 |
| Manes.02G199800 | TNL | 12,8 |
| Manes.10G121812 | TNL | 130,1 |
| Manes.10G126600 | TNL | 417,9 |
| Manes.10G126900 | TNL | 176,6 |
| Manes.10G127950 | TNL | 185,3 |
| Manes.10G127975 | TNL | 15,9 |
| Manes.10G128001 | TNL | 60,1 |
| Manes.14G148375 | TNL | 28,7 |
| Manes.18G112080 | TNL | 25,6 |

| TME3 67 dpi | | |
|---|---|---|
| Gene ID | Class | log FC |
| Manes.02G214775 | CNL | 4,7 |
| Manes.07G009201 | CNL | 4,2 |
| Manes.10G127500 | TNL | 5,4 |
| Manes.02G205700 | TNL | 6,3 |

Legend:
6
4
3
2
1
0
-1
-2
-3
-4
-5

**Fig 2. Differentially-expressed NLR genes in susceptible T200 and tolerant TME3 genotypes infected with SACMV. (a)** Total number of RNA transcripts DE in T200 and TME3 infected with SACMV at 12, 32 and 67 dpi. **(b–d)** A heatmap of DE NLR genes in T200 and TME3 infected plants at 12, 32 and 67 dpi. NL = nucleotide-binding LRR; RNL = coiled-coil NLR; CNL = RPW8-like coiled-coil NLR; and TNL = Toll/Interleukin-1 receptor/resistance NLRs. **(e)** Total number of NLR transcripts DE in T200 and TME3 infected with SACMV at 12, 32 and 67 dpi. **(f)** Overlap between DE genes in T200 and TME3 induced by SACMV at 12, 32 and 67 dpi. Values indicated in the Venn diagram present the number of downregulated and upregulated genes in T200 and TME3 compared to the mock infection. Transcripts were considered DE if the $\log_2$ fold-change > 0.5 and P ≤ 0.05. (n = 3 to 4 leaves collected from individual plants and one measurement per time point was taken and considered an overall representation of 3 biologicals). Numerical values in the heatmap indicate expression level calculated as F-value. Genes labelled in red and blue show significant upregulation and downregulation respectively in response to SACMV infection, compared to mock infection.

**Table 2. Expression levels of SACMV DNA-A and DNA-B encoded virulence factors in TME3 and T200 at 12, 32 and 67dpi measured by RT-qPCR.**

| Genotype & SACMV virulence factors | 2^dCq 12 dpi | 2^dCq 32 dpi | 2^dCq 67 dpi |
|---|---|---|---|
| **T200-AV1** | 55,289 ± 4,997 | 265,594 ± 24,705 | 10,540 ± 808 |
| **TME3-AV1** | 5,607 ± 1,784*** | 65,535 ± 5,363*** | 5,752 ± 1,194 |
| **T200-AC2** | 156,459 ± 31,289 | 4,601 ± 357 | 137 ± 13 |
| **TME3-AC2** | 4,928 ± 499** | 275 ± 64*** | 248 ± 35* |
| **T200-AC4** | 11 ± 3 | 41 ± 11 | 615 ± 91 |
| **TME3-AC4** | 14 ± 3 | 161 ± 14*** | 197 ± 29* |
| **T200-BC1** | 25 ± 9 | 36 ± 7 | 9,526 ± 348 |
| **TME3-BC1** | 17 ± 5 | 10,810 ± 56*** | 3,118 ± 555* |
| **T200-BV1** | 11 ± 3 | 51 ± 5 | 8,464 ± 1,362 |
| **TME3-BV1** | 14 ± 3 | 7,568 ± 871*** | 3,723 ± 183* |

Transcriptional activator protein (AC2); silencing suppressor protein (AC4); coat protein (AV1); movement protein (BC1) and nuclear shuttle protein, NSP (BV1). Data represent the mean of three independent biological replicates. Plants were grown side by side under identical conditions (n ≤ 5 leaves for each genotype per time point). Values represent averages ± SE. Significant difference: *P < 0.05; **P < 0.01; ***P < 0.001. *GTPb* was used as a housekeeping gene.

downregulated in TME3-symptomatic leaves, whereas ETI and SA key and indicator genes were upregulated (Figs 3d–g). Indeed, the NIK-pathway key and indicator genes were upregulated in asymptomatic-TME3 (recovered) leaves (S2 Fig).

Given that WRKYs are the transcriptional regulators of SA biosynthesis genes by binding directly to the promotor regions of the respective genes [79], the total WRKY transcripts from the transcriptome data were analysed in T200 and TME3 at 12, 32 and 67 dpi. WRKY expression was overall downregulated in susceptible T200 at all time points but in TME3, WRKYs were upregulated at 32 dpi (Fig 4a). These results indicate that WRKY-TFs may be involved in inducing tolerance by elevating the levels of SA in TME3. In addition, WRKY-TFs positively regulate ROS and together with SA promote pathogen resistance. In agreement with this, ROS levels were significantly lower at 32 dpi when both T200 and TME3 genotypes were highly symptomatic (Fig 4b). However, ROS levels were comparable between symptomatic and asymptomatic leaves at 67 dpi, a time point where TME3 leaves show tolerance. This can be linked in TME3 recovered leaves ( S2a Fig) to a significant expression of *RBOHD,* an NADPH oxidase that catalyses the production of superoxide ($O_2$-) which peroxides convert into $H_2O_2$ [80].

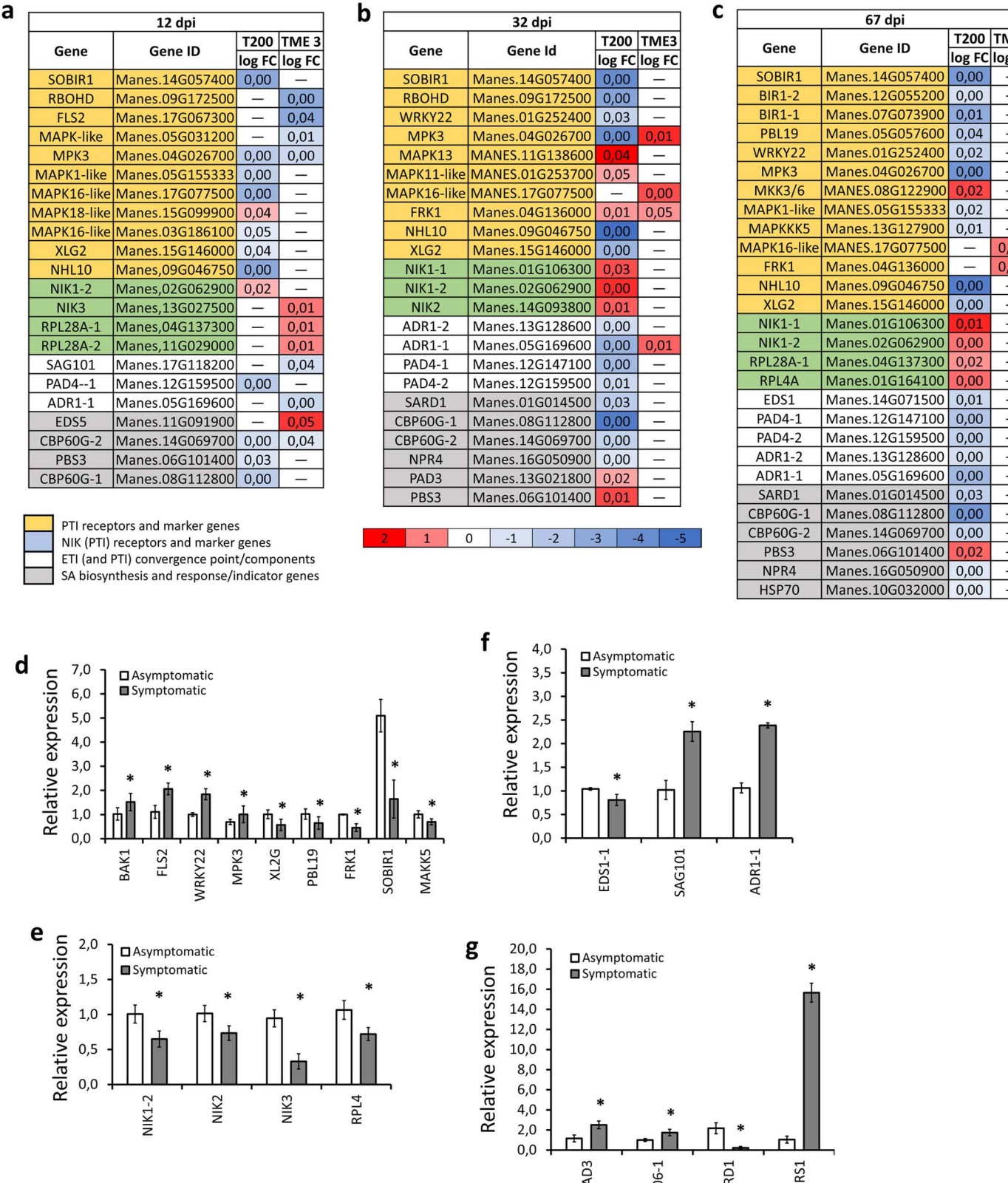

**Fig 3. Transcript levels of MAPK, NIK, NLR and SA-dependent gene pathways and responsive genes.** Heatmap represents $\log_2$-transformed fold change of PTI, ETI and SA transcripts DE in T200 and TME3 SACMV infected and mock leaves at **(a)** 12, **(b)** 32 and **(c)** 67 dpi. Transcripts were considered DE if the $\log_2$ fold-change

> 0.5 and P ≤ 0.05. Numerical values in the heatmap indicate expression level calculated as P-value. Genes labelled in red and blue show significant upregulation and downregulation respectively. (n = 3 to 4 leaves collected from individual plants and one measurement per time point was taken and considered an overall representation of 3 biologicals). RT-qPCR assay for PTI **(d)** MAPK, **(e)** NIK, **(f)** ETI and **(g)** SA key and response genes. Data represent the mean of three independent biological replicates. Error bars represent SD. An asterisk indicates a statistically significant difference according to unpaired Student's t-test (two-tailed), *p ≤ 0.05. *GTPb* was used as a housekeeping gene.

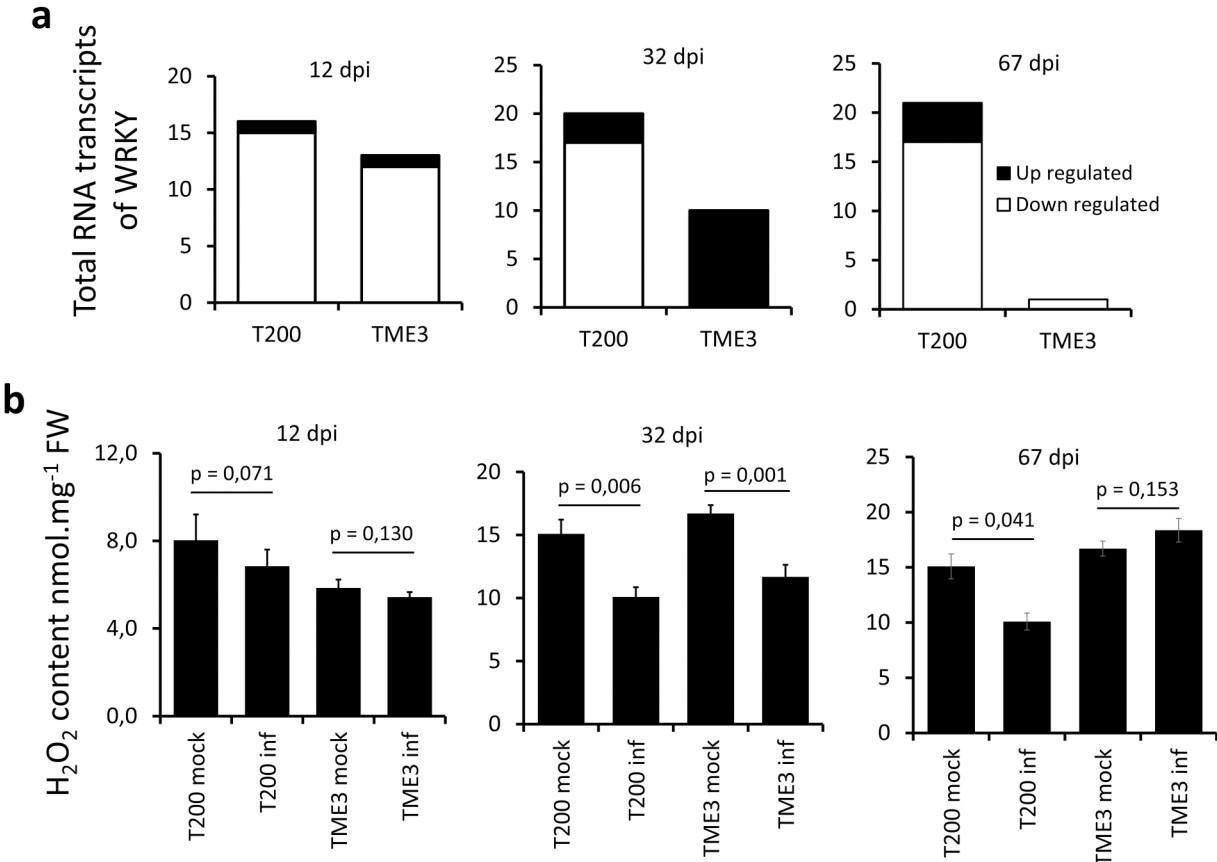

**Fig 4. Expression profile of WRKY-TFs and ROS accumulation in susceptible T200 and tolerant TME3 cassava genotypes.** Total number of all DE RNA WRKY-TF transcripts **(a)** and ROS accumulation in T200 and TME3 leaves infected with SACMV at 12, 32 and 67 dpi **(b)**. WRKY-TFs were considered DE if the log₂ fold-change > 0.5 and p ≤ 0.05. (n = 3 to 5 leaves from individual plants combined and considered an overall representation of least 3 biological replicates. ROS data is the mean and SE of four independent biological replicates; significant differences in p values were determined by one-way ANOVA).

## Nucleotide leucine-rich repeat and cell-surface nucleotide leucine-rich repeat receptor proteins/kinases are polymorphic

Having considered the expression profile of ETI, PTI and SA in response to SACMV infection in susceptible (T200) and tolerant TME3, the follow-up question was whether polymorphism in intracellular and/or cell-surface receptors in cassava genotypes of different CMD responses can explain susceptibility, resistance and/or tolerance to CMD. The coding regions of intracellular and cell-surface receptors in 24 cassava genotypes were analysed using PacBio sequencing (Table 3). The selected genes were considered to be a general representation of the cassava cell-surface LRR receptor kinases (LRR-RKs), cell-surface LRR receptor protein (LRR-RP) and intracellular nucleotide-binding receptors (NLRS). Allelic variation and structural mutations

**Table 3. Total count of HiFi reads indicating gene variation in 24 cassava genotypes.**

| Accession | HiFi Reads | SACMV response | *Me05G169600* RNL | *Me07G004200* LRR-RP | *Me10G127900* TNL | *Me11G060000* LRR-RP | *Me12G091600* NBS-LRR | *Me 11G122832* CNL |
|---|---|---|---|---|---|---|---|---|
| *M. glaziovii* | 5,710 | Resistant | 57 | 2,819 | 92/23 | 44 | No amplicon | No amplicon |
| Ukulinga 2 | 3,841 | Resistant | 12 | 1,912/353 | Fail | 32/15 | No amplicon | Fail |
| Ukulinga 8 | 5,632 | Susceptible | 52/18 | 1,226 | 39 | 23/12/10/8 | 1,715 | 619/101/17 |
| Ukulinga 9 | 2,718 | Susceptible | 21/12 | 381 | 11/7 | 7 | 779 | 77/37 |
| T200 | 7,535 | Susceptible | 271/226 | 4,144 | 98 | 14 | 1,217 | 21 |
| Slira 21 | 4,095 | Susceptible | 39/33 | 911 | 549 | 7 | 995 | 781 |
| P10/3 | 2,524 | Susceptible | 11/5 | 68/59/28 | 5 | 128 | 659 | 63 |
| P4/10 | 25,563 | Susceptible | 970 | 15,859 | 779 | 77/24 | No amplicon | No amplicon |
| P1/19 | 15,425 | Susceptible | 146/63/52 | 3,258 | 1,228 | 33/13/9 | 5,573 | 8 |
| SM1432.4 | 5,085 | Susceptible | 112 | 50/50 | 107 | 54/20 | 2,125 | 1,115 |
| SM707-17 | 8,667 | Tolerant | 25/14/11/9 | 4,929 | 66/26 | 17 | No amplicon | 26 |
| AR17.3 | 3,463 | Susceptible | 125/124 | 341/275/81 | 18 | 216 | 341 | 22 |
| AR40.17 | 4,602 | susceptible | 150/124/106 | 582/523/185 | 87/37/18 | 48/21 | 480 | 18/ 15 |
| CR43.3 | 5,138 | Resistant | 272/231/102 | 141/70/51 | 90 | 16/7 | No amplicon | 39 |
| cv.60444 | 3,708 | Susceptible | 108 | 1,592 | 628 | 11/8/5 | 308 | No amplicon |
| TME 1 | 4,978 | Resistant | 20/15 | 2,938 | 201 | 8 | 179 | 17 |
| TME 3 | 3,833 | Tolerant | 254/242 | 1,828 | 51 | 8 | 360 | 15 |
| TME 117 | 1,139 | Susceptible | 33 | 67/43/20/17 | Fail | Fail | 216 | 91/84 |
| TMS98/0002 | 1,563 | Resistant | 16/9/7 | 175/142/94 | 67 | 18/16 | 101/59/16/10/8/7 | 5 |
| TMS98/0505 | 2,374 | Resistant | 55/22 | 116/98/81 | 89/78/60/38/ | 6 | Fail | 19 |
| TME419 | 2,011 | Resistant | 35/50 | 92/66/51 | 86 | Fail | 35/29/11/5 | 72/53 |
| IBA01/1368 | 2,540 | Resistant | 72/61 | 557 | 29 | 41/12 | 111 | Fail |
| IBA96/1632 | 7,764 | Tolerant | 47/43/28 | 3,497 | Fail | 53 | 746 | 163 |
| IBA95/0289 | 2,513 | No tested | 33 | 469/345/207/154 | 16/8/8 | 11 | 102 | 5 |

No amplicons = no PCR product(s) i.e. the amplicons do not match the cassava reference genome AM250-2 genes; Fail = PacBio seq failed.

in the coding regions were present in the analysed genes (Table 3). In some genotypes, gene structural variation was high and no PCR amplicons were obtained, or the HiFi reads could not be resolved based on the AM250-6 reference genome. These results showed that the RPW8 domain of cassava ADR1 protein (Manes.05G169600) was the most conserved and had two amino acid synonymous mutation substitutions, whereas there were multiple polymorphic regions in the NBS-LRR domains (S2 Fig). The NBS-LRR and CC-NBS-LRR domains of Me12G091600 and Me11G122832, respectively, had minimal amino acid variation compared to the other sequenced R genes, yet PCR amplicons for these two NLR genes failed in many genotypes, suggesting mutation(s) that prevent PCR amplification of these genes. The TNL (*Me10G127900*) and two LRR-RPs (*Me07G004200* and *Me11G060000*) genes were highly variable, and the mutation appeared to occur at random. These results indicate that *Manihot esculenta* LRR-RK, LRR-RP and NLR genes are polymorphic across genotypes and these results were consistent with the observations that LRR-RP and NLR genes are polymorphic with LRR-RK-encoding genes being more conserved [81].

Given that LRR-RKs/LRR-RPs and NLRs are polymorphic [81], the question was, do LRR-RKs/LRR-RPs and NLR polymorphisms segregate with susceptibility, tolerance or resistance to SACMV. For example, seventy percent of the cassava genotypes examined had more than two alleles of the *Me.ADR1* gene (*Manes.05G169600* and a similar trend was observed with

other genes analyzed (Table 3)). The interpretation of gene heterozygosity in individual cassava genotypes was a result of (1) gene duplication of NLRs and LRR-RKs/LRR-RPs within the genus and individual genotypes and (2) gene duplication due to polyploidy occurrence. Furthermore, there was no correlation between particular NLR resistance gene mutation(s) and immune/susceptible/tolerant responses to SACMV. Additionally, there was no link with *CMD1*, *CMD2,* and *CMD3* loci, and the pattern of gene polymorphism can be better explained by phylogeny relationship (and breeding history).

## Discussion

Resistance to cassava mosaic geminiviruses is linked to *CMD1*, *CMD2* and *CMD3* loci, and is characterised by low viral accumulation and/or incomplete systemic spread of cassava mosaic geminivirus [82]. However, there was no apparent correlation between genotypes displaying disease, immune or tolerant responses and the presence of these three loci (Table 3). Furthermore, the molecular mechanisms and role of SACMV virulence factors involved in CMD responses are poorly characterized. Geminiviruses recruit host factors for replication and movement, and all geminivirus-encoded proteins may potentially act as virulence factors [83]. Notably in this study, SACMV *AV1* and *AC2* virulence genes appear to play a significant role in systemic infection in cassava, as the establishment of successful infection in T200 and TME3 is likely dependent on early expression of virulence factors *AC2* and *AV1* at 12 dpi. At 67 dpi, expression of SACMV virulence gene transcripts, *AC4, BC1,* and *BV1* increases in both genotypes while *AC2* and *AV1* decrease, suggesting a switch in SACMV virulence factors between early and late infection stages to maintain viral infection. The geminivirus AC4 protein has multiple overlapping functions. Transcription of SACMV *AC4* virus suppressor (VSR) of host antiviral PTGS, increased from 12 to 67 dpi as CMD progressed in both T200 and TME3. However, notably in TME3 at recovery, *AC4* expression was significantly lower compared to T200 where expression of *AC4* increased by 3-fold (Table 2). It is worth noting that while *AC4* was expressed at low levels in both T200 and TME3 compared to some other virulence factors, it was significantly upregulated in TME3 at 32 dpi compared to T200. However, due to very low *AC4* transcript levels compared to *AC2* and *BV1* transcripts (Table 2), it is very unlikely that AC4 plays a highly significant pathogenicity role during SACMV infection in cassava.

It was concluded that once SACMV establishes infection, BV1-encoded NSPs, and to some extent BC1-encoded MPs, act as the main pathogenic determinant factors later during virus intercellular spread/movement (67 dpi). Moreover, while *NSP (BV1)* transcripts increased 166-fold at 67 dpi compared to 32 dpi in T200, transcripts were reduced by 2-fold at 67 dpi in TME3 recovered leaves. Since NSPs are responsible for delivery of DNA-A and DNA-B to the nucleus for replication, suppression of NSPs during begomovirus is a key determinant of tolerance in the west African TME3 germplasm.

Additionally, this study proposes that during (1) early infection, the TrAP protein (AC2) establishes virus infection by inhibiting host PTGS, and (2) the NSP becomes the main pathogenic determinant at the late stage of infection (SACMV DNA-A and SACMV-B replicate in the nucleus) by suppressing PTI responses (Fig 5a). The tolerance/recovery phenomenon is achieved as a result of suppressed viral NSP gene expression and activation of ROS dependent basal immune response via the NIK-pathway, and possibly TNL-mediated immune responses (Fig 5b). These results indicate that gains in limiting the accumulation of SACMV virulence factors contribute to tolerance to SACMV. This hypothesis is further supported by AC2 of both *East African cassava mosaic virus (*EACMV) and *Indian cassava mosaic virus* (ICMV), demonstrated to be a suppressor of PTGS [28]. Significantly, EACMCV and ICMV induce

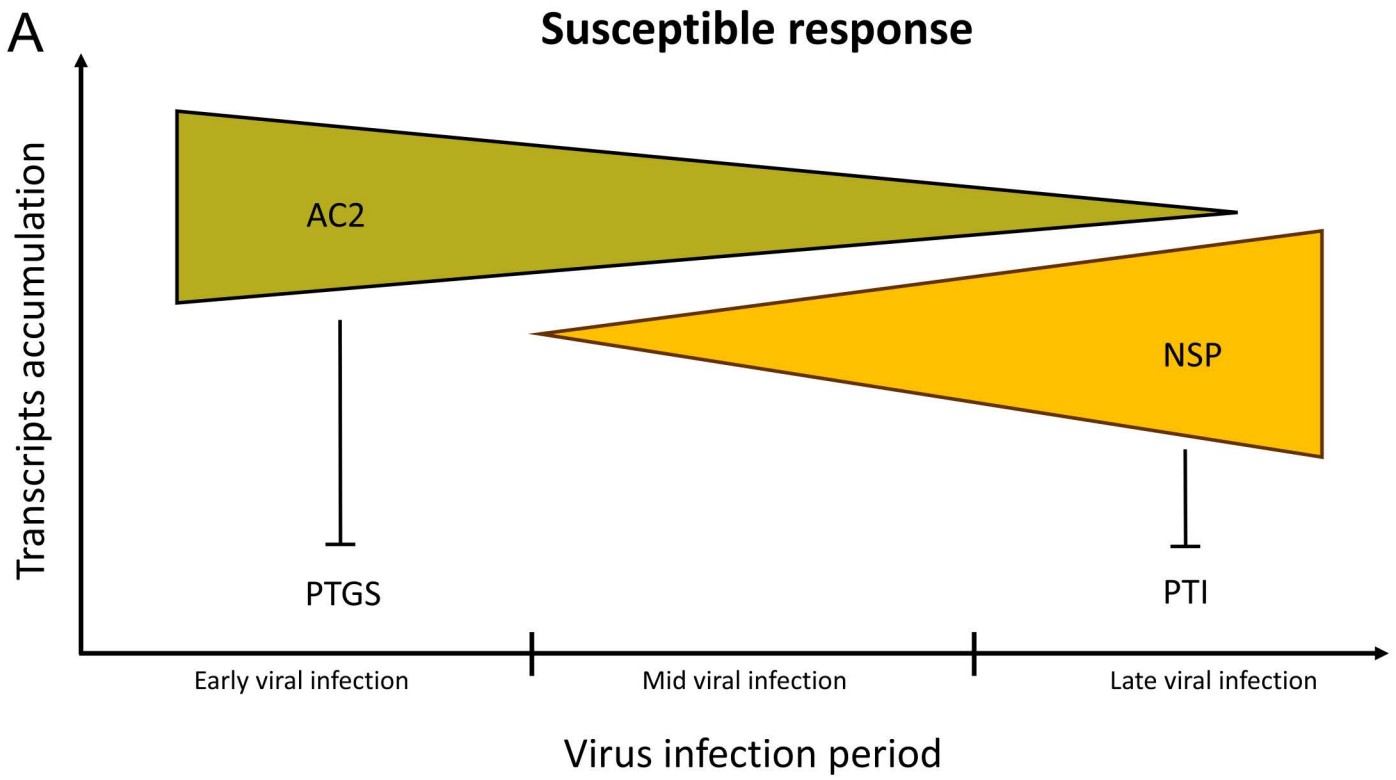

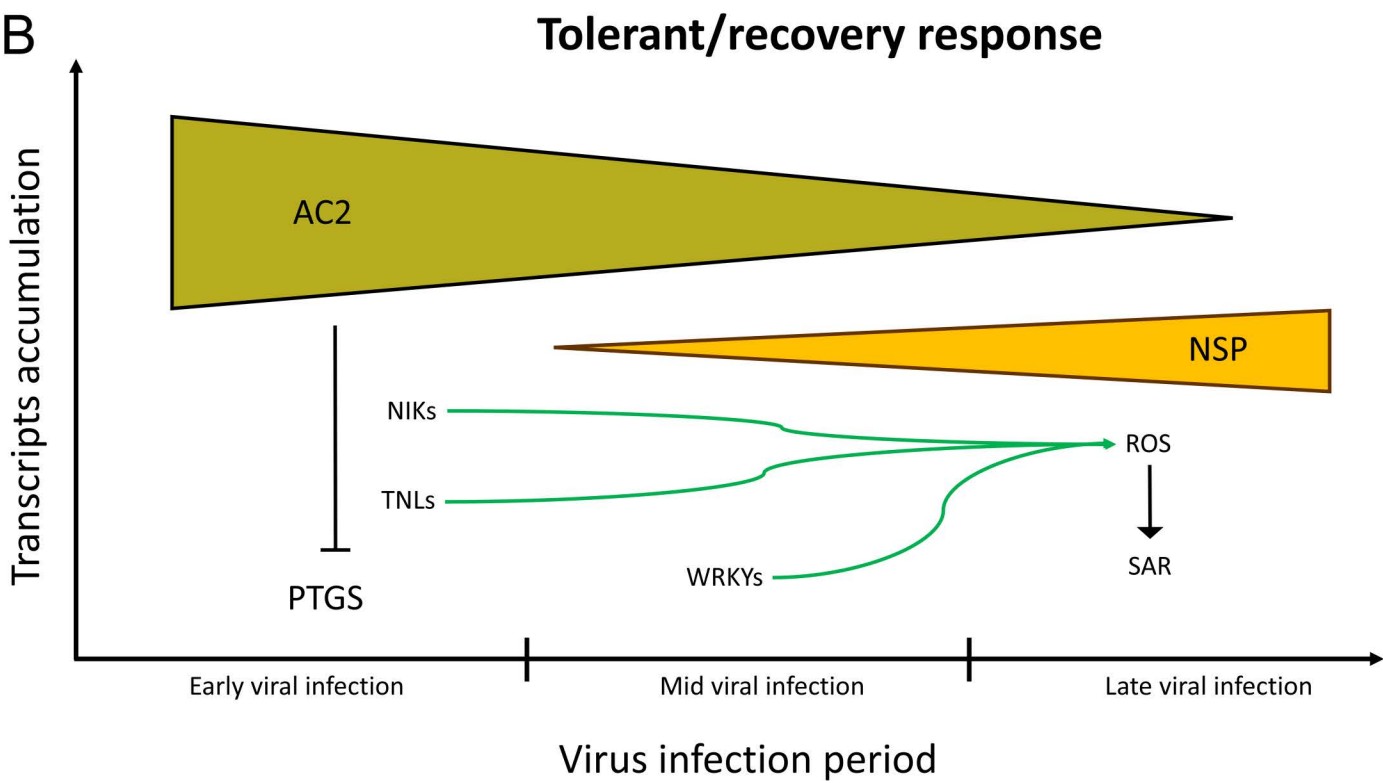

**Fig 5. Hypothetical model of the molecular mechanism underlying the ability of SACMV to suppress the host RNA silencing mechanism (PTGS) and systemic acquired resistance (SAR).** In susceptible response, SACMV AC2 encodes the transcriptional activator protein (TrAP) and suppresses host PTGS in both susceptible

response (a) and tolerance/recovery response **(b).** During the late infection stage, suppression of host PTGS by SACMV is alleviated and nuclear shuttle protein (NSP) encoded by *BV1* to promote virus cell-to-cell movement and inhibition of PTI in susceptible T200 (susceptible response) but the expression of NSP is suppressed in the tolerant/recovery response (TME3) possibly by NIKs, TNLs and WRKYs function through elevation of ROS levels and activation of SAR. PTGS: posttranscriptional gene silencing; NSP: nuclear shuttle protein; NIK: NSP-interacting kinase 1; SAR: systemic acquired resistance; ROS: Reactive oxygen species; TNL: N-terminal Toll/interleukin-1 receptor (TIR) domains; PTI: pathogen-triggered immunity.

PTGS even though plants do not gain tolerance, illustrating that activation of only PTGS might not be sufficient to suppress virus replication [28]. These studies and herein suggest a crosstalk between PTGS and PTI in plant antiviral responses.

The high expression at 12 dpi of *AV1* which encodes the coat protein (CP), that also functions in cell-to-cell movement and systematic spread of the virus, can be associated with high expression levels of *AC2*. *AC2* is a primary positive regulator (activator) of *AV1* and other viral genes [84], moreover, the AC2 protein is a suppressor of host gene silencing antiviral defence mechanism [27,85]. High expression of SACMV AC2 transcripts in T200, particularly during early (12 dpi) infection, would mediate suppression of PTGS, TGS, and other regulatory genes, to establish SACMV infection. However, a close examination of expression levels of SACMV DNA-A and DNA-B encoded virulence factors during plant systemic infection (at 32 dpi), showed an overrepresentation of DNA-A encoded virulence factors in T200, in contrast to DNA-B encoded virulence factors in TME3 (Table 2). These results demonstrate that the expression of SACMV virulence factors is dependent on a host genotype. Differences in TME3 and T200 responses to SACMV have also been shown to be a result of several other host responses, such as PTGS [39,40], demonstrating the complexity and multifactorial nature of plant responses to geminivirus infection.

The high accumulation of the MP encoded by BC1 aligns with increased NSP encoded by BV1 transcripts at 32 dpi in TME3, suggesting that SACMV moves efficiently from cell-to-cell and into the nucleus in this cultivar in order to trigger subsequent tolerance timeously. Notably, at the subsequent recovery phase (67 dpi), expression of both MPs decline only in TME3, leading to a decrease in SACMV replication, which is regarded as an important signature for tolerance and recovery in TME3 [67]. In contrast, replication and movement of SACMV increase at 67 dpi in T200 (Table 2). Intriguingly, the MP and not NSP of tomato leaf curl New Delhi virus (ToLCNDV) is necessary for transmission by whiteflies, such as *B. tabaci* [86], nevertheless, the mode of transmission differs within ToLCNDV isolates. The MPs are required for cell-to-cell and long-distance movements and are believed to form a complex with the NSP-bounded viral DNA and transport viral DNA through the plasmodesmata for phloem-limited begomoviruses. For mesophyll-invading begomoviruses, MPs take over viral DNA from NSP in the cytoplasm and transfer it to adjacent and distant cells. A rapid increase in expression of *BC1* and *BV1* virulence factors at 67 dpi in T200 correlates with persistent CMD symptoms and SACMV replication [67].

Moreover, the NSP subverts plant antiviral immunity through protein-protein interactions with LRR-RLK receptors [32]. To date, NUCLEAR SHUTTLE PROTEIN-INTERACTING KINASE 1 (NIK1), is the only LRR-RLK that induces PTI through physical interaction with the geminivirus NSP protein [58]. BV1-encoded *NSP* and *AC4* transcripts increased concomitantly at 67 dpi in susceptible T200, while they declined in recovered TME3 leaves. Taken together, these results suggest a synergic mechanism of action between AC4 and NSP to establish late viral infection in cassava. On the other hand, expression of viral *AC2* transcripts was significantly high at early infection suggesting that AC2 plays an important role in early SACMV systemic infection.

Transcriptome results showed that SACMV induced a higher transcriptional response in susceptible T200 compared to tolerant TME3 at all time points post-infection, and the number of NLR-PTI-SA differentially expressed genes gradually increased over the infection period in T200 whereas in TME3 they decreased. While SACMV replication was considerable at 32 dpi in TME3, it was significantly lower than T200, indicating that in TME3 SACMV replication was effectively restricted prior to 32 dpi. Given that 12 dpi is the time point where TME3 is transcriptionally active in response to SACMV infection, the overall downregulation of NLR receptor proteins supports further that the tolerance response may dependent on other mechanisms such as PTGS. Nonetheless, four TNLs were upregulated in TME3 whereas all the TNLs were downregulated in T200 at 12 dpi, and these four TNLs may play a role in SACMV early immune response. TNLs-based immunity corresponds to a rapid induction of TNLs in response to diverse PAMPs [87], and TNLs are necessary for reinforcing the plant immunity response [88]. It is also suggested that an oxidative burst may play a role in tolerance to cassava mosaic begomoviruses. The oxidative burst is a rapid, transient, production of huge amounts of reactive oxygen species (ROS) and is one of the earliest observable aspects of a plant's defence strategy. Ngou et al. [42] and Yuan et al. [89] reported that NLRs boost the PAMP-triggered ROS ($H_2O_2$) burst in the apoplast, leading to ROS accumulation to levels higher than PTI or ETI alone. An identical ETI-PTI crosstalk against geminiviruses in recovery response (tolerant phenotype) is probable and the downregulation of a membrane-localized *NADPH/respiratory burst oxidase protein D* (*RBOHD*) is in supports this hypothesis (S1 Fig). The CP of *Plum pox virus* [56] and MPs of *Cucumber mosaic virus* [55] were reported to suppress flg22-induced ROS production. Therefore, the downregulation of *FLS2* and *RBOHD* in TME3 at 12 dpi (Fig 3a) indicates that SACMV suppresses the ROS burst. Similar to RBOHD, SA-dependent and SA-independent basal defence pathways were downregulated during the systemic viral spread. SA can directly induce antiviral defence through its effect on ROS generation in the mitochondria. TNLs can boost immunity signalling by inducing the accumulation of salicylic acid (SA) hormones [90]. Therefore, TNLs may boost immune signalling involving SA biosynthesis and ROS burst in TME3 at 12 dpi.

This study also suggests a possible additive and synergistic effect of NIKs (*NIK1/NIK2/NIK3)* antiviral immunity through induction of global translational suppression and possible MAPK activation. RT-PCR showed a downregulation of PTI (MAPKs and NIKs) key and response genes whereas ETI and SA key and response genes were upregulated at 32 dpi symptomatic leaves. In general, *NIK1* (and to a lesser extent *NIK2*) is a negative regulator of the flg22-induced ROS burst, with *NIK3* believed to be an antagonist to *NIK1* and *NIK2* [58]. In addition, upon begomovirus infection, NIK1 induced global translation suppression as a plant antiviral immunity mechanism [91]. MAPK cascades are a convergent point in both PRR- and NLR-immune-mediated signalling [92]. These findings indicate that individual immune mechanisms to SACMV infection equally contribute to cassava tolerance to geminiviruses.

A search for the genes responsible for *CMD2*-type resistance (genes within the CMD2 locus) in cassava found two peroxidase genes associated with CMD severity [15,16,20]. Interestingly, one of the identified peroxidases was shown to have three nonsynonymous mutations that were specific to the susceptible genotypes [15] and possibly perturb endogenic ROS production. Interestingly, Ngou et al. [42] and Yuan et al. [89] reported that PAMP recognition leads to an activated *RBOHD* resulting in the production of extracellular ROS. Leaves of recovered TME3 plants showed an upregulation of *RBOHD* and high levels of $H_2O_2$ compared to symptomatic plants at 67 dpi. In accordance with this, one can conclude that nonsynonymous mutations perturbed the peroxidase function resulting in lower $H_2O_2$ production in non-recovering plants. Indeed, priming cassava stem cuts before planting with $H_2O_2$ reduces

cassava mosaic disease incidence by 70% [93]. Moreover, WKRYs expression is linked to high production of ROS, by directly activating the expression of *RBOHD* [94]. Our results suggest that NIK signal pathway and production of $H_2O_2$ are necessary for cassava-acquired immunity to geminiviruses.

Analysis of NLRs diversity and polymorphism in cassava genotypes showed that NLRs are highly polymorphic. However, ADR1, a NBS-LRR disease resistance protein that possesses N-terminal kinase subdomains, is a PTI-ETI-SA convergent point and induces basal immunity, had low gene variation with the N-terminus domain (CC-) showing no polymorphism indicating that the PTI-ETI-SA convergent point is highly conserved between cassava SACMV phenotypic responses. However, having shown that NLRs are polymorphic, there was no correlation between NLR gene polymorphism and SACMV responses in 24 tested cassava genotypes.

## Conclusions

The results in this study are in line with other previous reports in T200 and TME3 that demonstrate that the different responses of cassava genotypes to geminiviruses are dependent on viral accumulation and recovery (tolerance), and that resistance depends on the ability of a plant to restrict or suppress virus replication. The differential response to SACMV in different cassava genotypes underlines there are several molecular mechanisms at play in how cassava responds to geminivirus infection. This study indicated that at the early infection stage (12 dpi) both susceptible T200 and tolerant TME3 already perceived SACMV-derived effector(s), but that those in TME3 may have led to an induced effector-triggered immunity (ETI) response, whereas in T200 the induced NLRs were likely associated with effector triggered susceptibility (ETS). Whether exhibiting susceptibility or tolerance, it was evident that many NLR genes (53%) expressed in TME3 were also expressed in T200, indicating that a common ETI pathway was induced in the susceptible T200 and tolerant TME3 responses. Thus, these results indicate that there is a crosstalk between tolerant and susceptible responses to geminivirus infections in cassava.

Secondly, this study proposes that SACMV suppresses MAPK and NIK-mediated immune response leading to a successful viral infection. NIK-mediated antiviral immunity confers resistance to begomoviruses through suppression of global translation. Consistent with this, in this study *NIK3* and small ribosomal proteins (*RPL28A* and *RPS13B*) were upregulated in the tolerant plants. *NIK1* and *NIK2* were expressed at 32 dpi and 67 dpi in T200 with distinct small ribosomal proteins compared to TME3. Therefore, the conclusion is *NIK3* contributes to the suppression of SACMV replication, while *NIK1* and *NIK2* do not contribute to viral tolerance. These results suggest NIK3-mediate antiviral response in the early days of infection and geminiviruses reinforce antiviral suppression through inactivation of MAPKs and ETI and SA signalling. Future gene editing or genetic engineering related to NIK3/PTI in farmer-preferred or commercially grown cassava cultivars could improve tolerance/basal immunity to CMD and consequently starch yield.

## Supporting information

**S1 Table. CMD-type resistance and cassava genotypes symptomatic scoring determined using a scale of 1–5.** SI refers to SACMV symptom index.
(DOCX)

**S1 Fig. Transcript levels of MAPK, NIK, NLR and SA-depended gene pathways and responsive genes.** RT-qPCR assay to validate the transcriptome data for PTI (MAPK **(a)** and NIK **(b)**), ETI **(c)** and SA **(d)** key and response genes diff expressed in asymptomatic and

recovered TME3 leaves. Data represent the mean of three independent biological replicates. Error bars represent SD. An asterisk indicates a statistically significant difference according to unpaired Student's t-test (two-tailed), *p ≤ 0.05. *GTPb* was used as a housekeeping gene. (n = 3 to 5 leaves from individual plants combined were measured, one measurement per time point.
(TIF)

**S2 Fig. Structure-based alignment and homology modelling of MeTIR domains. (a)** AM560-2 TIR domains have conserved glutamic acid (indicated by a yellow triangle) required for NADase activity. Cylinders above the alignment and arrows indicate conserved secondary structure residues generated from RUN$^{TIR}$ (PDB ID: 7rx1), RPV1 (PDB ID: 5ku7) and RPP1 (PDB ID: 5teb) using PROMALS3D, Grishin Lab. **(b)** Homology modelling of cassava TIR domains differentially regulated in TME3 at 12 dpi using Manes.18G125400 as a model. The yellow triangle on the bottom indicates the conserved glutamic acid required for NADase activity and 2',3'-cAMP/cGMP synthetase and the blue cycle indicates conserved cysteine important for the cleavage of nucleic acids and synthesis of 2',3'-cNMPs.
(TIF)

## Acknowledgments

We thank the International Institute of Tropical Agriculture (IITA), Nigeria and the Agricultural Research Council (ARC), South Africa for providing cassava germplasm.

## Author contributions

**Conceptualization:** Bulelani L. Sizani, Marie E.C. Rey.

**Data curation:** Bulelani L. Sizani, Keelan Krinsky, Oboikanyo A. Mokoka.

**Formal analysis:** Bulelani L. Sizani, Keelan Krinsky.

**Funding acquisition:** Marie E.C. Rey.

**Investigation:** Bulelani L. Sizani, Keelan Krinsky, Oboikanyo A. Mokoka.

**Methodology:** Bulelani L. Sizani, Oboikanyo A. Mokoka.

**Project administration:** Marie E.C. Rey.

**Resources:** Marie E.C. Rey.

**Supervision:** Marie E.C. Rey.

**Writing – original draft:** Bulelani L. Sizani.

**Writing – review & editing:** Marie E.C. Rey.

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
