## [Decision Letter · Decision Letter 0]

20 Aug 2024

PONE-D-24-29673Comparative analysis of pattern-triggered and effector-triggered immunity gene expression in susceptible and tolerant cassava genotypes following begomovirus infectionPLOS ONE

Dear Dr. Rey,

Thank you for submitting your manuscript to PLOS ONE. After careful consideration, we feel that it has merit but does not fully meet PLOS ONE’s publication criteria as it currently stands. Therefore, we invite you to submit a revised version of the manuscript that addresses the points raised during the review process.

We look forward to receiving your revised manuscript.

Kind regards,

S.V. Ramesh, PhD

Academic Editor

PLOS ONE

Journal Requirements:

6. Please amend your manuscript to include your abstract after the title page.

Additional Editor Comments:

This manuscript is technically sound and a valuable piece of work that merits publication. However, the peer reviewers suggest that the authors revise the manuscript. If you choose to revise it, please address all the minor queries posed by the reviewers and resubmit the manuscript for further consideration.

Reviewers' comments:

Reviewer's Responses to Questions

**Comments to the Author**

1. Is the manuscript technically sound, and do the data support the conclusions?

Reviewer #1: Yes

Reviewer #2: Partly

2. Has the statistical analysis been performed appropriately and rigorously? 

Reviewer #1: Yes

Reviewer #2: I Don't Know

3. Have the authors made all data underlying the findings in their manuscript fully available?

Reviewer #1: Yes

Reviewer #2: No

4. Is the manuscript presented in an intelligible fashion and written in standard English?

Reviewer #1: Yes

Reviewer #2: Yes

5. Review Comments to the Author

Reviewer #1: The manuscript ‘Comparative analysis of pattern-triggered and effector-triggered immunity gene expression in susceptible and tolerant cassava genotypes following begomovirus infection’ [PONE-D-24-29673] is well-conceptualized, well-planned, and well-written. However, I have a few suggestions which could improve the manuscript.

Please provide key quantifiable results in the abstract section.

Line 14: ‘nearly a billion’ please quantify with the recent FAO database.

In the introduction, please mention various types of CMV and discuss the virulence of SACMV with SLCMV (Sri Lanka) and NDCMV (New Delhi).

Line 60: Please delete ‘2019)’

Line 117: MAMPs/PAMPs also induce Callose deposition, and cell death in leaves in addition to ROS. Please mention.

Line 172: In plant-virus interactions.. please delete ‘some’

The introduction seems to be incomplete/abruptly concluded. Please mention the future implications of the expected outcome.

Moreover, the authors have introduced challenge inoculation at 12, 32, and 67 dpi and mentioned 12 dpi as an early stage. Is 12 dpi an early event? I doubt the author will miss early events such as ROS, cell death/HR, and callose deposition at 12 dpi. Please justify.

Please mention ‘g/L’ as ‘g L-1’ throughout the text.

Line 189: Whether the nodal explants were virus-indexed?

Line 227: Please mention the date of access for the software used in the MS.

Line 238-239: Please mention the make, city, and country of the consumables/kits/equipment used in this study.

Line 244: Please mention H2O2 instead of ROS.

Please mention H2O2 as the section 2.4 and Transcriptional analysis as 2.5 followed by library preparation (2.6)

Line 256: Please write the full form of SMRT in its first appearance

Please keep the list of primers Table S2 and Table S1 in the main text.

Line 275,276, 278, 279: Please mention the date of access for the software used in the MS.

Line 298: Please mention the Caption after the figure. (For all figures)

Line 380: Please delete the word ‘Footnote’ from all the tables. Please write the footnote below the Tables.

Line 384: *P < 0.05 or *P ≤0.05?

Line 478: Please revise (Table 2))

Results and discussion are adequate

Please mention a line of future implications of the output in the conclusion.

Please cross-check the language, grammar, and punctuation.

Please cross-check the reference pattern as per the journal standard.

Line 688: 8. 0 References…please check the section numbering per journal pattern.

The number of references (86) may be reduced.

Please use high-resolution images in the Figures/graphs.

The manuscript is well-prepared per the journal standard and may be considered with minor revision.

Good luck with the revision.

Reviewer #2: The author discussed the physiological changes in plants when inoculated with South African cassava mosaic virus (SACMV) using susceptible and non-susceptible cassava varieties.

In particular, this paper discussed the plant immune response that occurs a mutual potentiation of cell membrane receptor-associated pattern-triggered immunity (PTI) and nucleotide leucine-rich repeat (NLR) effector-associated immunity (ETI) in plant immune responses when inoculated with SACMV.

It is interesting that the amplification process of SLCMV was investigated in detail at each day after inoculation, physiological investigations such as quantification of ROS were also conducted, and the discussion was performed from various perspectives.

However, although the gene at the CMD2 locus, which functions most effectively in SLCMV resistance, has been identified in recent years, there is a lack of discussion on the relationship between mutations on CMD2 locus and the level of SLCMV symptoms.

The author needs to discuss two peroxidases near the CMD2 locus in the discussion section. Do these two types of gene changes occur during SLCMV infection?

Also, are there any special SNPs in the genes near the CMD2 locus? I think a partial revision of the Discussion section is necessary.

Regarding the quantitative of SMARTsequencing,

The number of HiFi Reads in Table 2 varies greatly among samples. Please explain whether this data reflects the amount of transcription and what statistical processing was performed.

Please increase the resolution of the figures. The text in Figures 2b, 2c, and 2d, and Figs. 3a, 3b, and 3c are difficult to see.

Please show the information of original or raw data of SMART sequencing obtained from Gene Expression Omnibus etc.

6. PLOS authors have the option to publish the peer review history of their article (what does this mean? ). If published, this will include your full peer review and any attached files.

**Do you want your identity to be public for this peer review?** For information about this choice, including consent withdrawal, please see our Privacy Policy .

Reviewer #1: **Yes: ** Manas R. Sahoo

Reviewer #2: No

---

## [Author Response · Author response to Decision Letter 1]

6 Dec 2024

RESPONSE TO EDITOR & REVIEWERS

Reviewer #1: The manuscript ‘Comparative analysis of pattern-triggered and effector-triggered immunity gene expression in susceptible and tolerant cassava genotypes following begomovirus infection’ [PONE-D-24-29673] is well-conceptualized, well-planned, and well-written. However, I have a few suggestions which could improve the manuscript.

Please provide key quantifiable results in the abstract section.

Response: Key significant quantifiable results (fold changes) were included for virulence factors AV1 and AC2 at 12 dpi in abstract (lines 12-18).

Line 14: ‘nearly a billion’ please quantify with the recent FAO database (year).

Response: Added reference, FAO, Save and grow: Cassava, a guide to sustainable production intensification. Rome: Organization of The United Nations, 2013 (line 35).

In the introduction, please mention various types of CMV and discuss the virulence of SACMV with SLCMV (Sri Lanka) and NDCMV (New Delhi).

Response: This below has been included (lines 62-69):

Cassava mosaic begomoviruses comprise 11 species, with nine of them found in Africa shown to be recombinants [12]. Indian cassava mosaic virus (ICMV) and Sri Lankan cassava mosaic virus (SLCMV) are found in Asia with SLCMV being more virulent than ICMV [14]. It is interesting to note that SLCMV has characteristics of a monopartite begomovirus. For example, SLCMV can produce systemic infection by interacting with DNAβ component of Ageratum yellow vein virus (AYVV) to produce systemic infection in Ageratum conyzoides, and SLCMV DNA-A alone causes systemic infection resulting in upward leaf curling and stunting in Nicotiana benthamiana [14].

Line 60: Please delete ‘(2019)’

Response: Deleted

Line 117: MAMPs/PAMPs also induce callose deposition, and cell death in leaves in addition to ROS. Please mention.

Response: Added: ‘MAMPs/PAMPs also induce callose deposition and cell death in leaves under biotic stress’ (line 160).

Line 172: In plant-virus interactions. please delete ‘some’

Response: Corrected: deleted (226).

Introduction

The introduction seems to be incomplete/abruptly concluded. Please mention the future implications of the expected outcome.

Response: Added (lines 213-224): An understanding of plant-virus interactions and dynamics is therefore critical for the development of strategies to manage the host response to virus infection, as well as in future to predict epidemics expected to escalate due to higher global temperatures. Begomovirus pandemics in central and East Africa have been linked to increased abundance of its whitefly insect vector, Bemisia tabaci Genn. which is also correlated with historical climate change [66]. Loss of cassava productivity due to begomovirus infection and increased whitefly vector populations, together with other effects of climate change, are having a substantial negative impact on the food security of mostly smallholder farmers in Africa. An increase in temperature, improving climate suitability for replication of whitefly populations and subsequent increased virus transmission, has already been observed in central East Africa [66]. Such environmental pressures in turn can drive viral pathogenesis. Understanding the impact of elevated temperature on the molecular, cellular, physiological, and epidemiological dynamics of the cassava-infecting begomoviruses and their cassava host is therefore crucial to mitigate the effect of climate change on this pathosystem in Africa.

Reviewer Comment: Moreover, the authors have introduced SACMV inoculation at 12, 32, and 67 dpi and 12 dpi is depicted as an early stage of infection of the plant. Is 12 dpi an early event?

I doubt the author will miss early events such as ROS, cell death/HR, and callose deposition at 12 dpi. Please clarify.

Response: To clarify in our system in which all our studies are done the cassava plantlets are grown up from tissue culture and potted in soil to acclimatize. At the 2-4 leaf stage the leaves are agro-inoculated with the SACMV virus infectious clones. The plants are only inoculated ONCE. At 12 days post inoculation (dpi) is considered early infection when we do the first sample. This is defined as when very mild symptoms just start appearing since the virus has been replicating and moving intercellularly/systemically. All our previous studies have shown that during the early infection stage (around 12 dpi), some response to begomovirus infection by cassava is occurring despite no visible CMD symptoms. SACMV has already begun to replicate, and while levels are low, already targets host PTGS (S J Rogans, F Allie, J E Tirant, M E C Rey (2016) Small RNA and Methylation Responses in Susceptible and Tolerant Landraces of Cassava Infected with South African cassava mosaic virus. Virus Research 225: 10–22). As infection progresses, symptoms of yellow mosaic and sometimes leaf curl appear/increase and are clear at 32 dpi which we use as our second time point and virus levels are high. Around 50 dpi and onwards to 67 dpi, replication in the tolerant TME3 cultivar begins to decrease and also symptoms attenuate (tolerance landrace NOT resistance), but in the susceptible T200 replication and symptoms continues to be high.

NOTE: We did not test for callose but H2O2 was measured at the three time points post inoculation.

Also, the perennial crop cassava does not show the typical HR/cell death in response to GVs at any stage unlike other model crops. When replication is high and virus is spreading systemically throughout the plant, yellow mosaic and some leaf curl are the typical symptoms. While a negative effect of SACMV on ROS production was observed early on at 12 dpi (Figure 4b), this was not significantly different between both susceptible (T200) and tolerant/recovery (TME3) cassava genotypes and healthy leaves, at this time point.

Please mention ‘g/L’ as ‘g L-1’ throughout the text.

Response: Corrected.

Line 189: Whether the nodal explants were virus-indexed?

Response: Yes. Added ‘virus-free cassava genotypes’

Line 227: Please mention the date of access for the software used in the MS.

Response: Corrected throughout MS.

Line 238-239: Please mention the make, city, and country of the consumables/kits/equipment used in this study.

Corrected throughout the MS.

Line 239: Please mention H2O2 instead of ROS

Corrected (line 280).

Please mention H2O2 as the section 2.4 and Transcriptional analysis as 2.5 followed by library preparation (2.6)

Corrected (line 315).

Line 256: Please write the full form of SMRT in its first appearance

Corrected

Please keep the list of primers Table S2 and Table S1 in the main text.

Corrected

Line 275,276, 278, 279: Please mention the date of access for the software used in the MS.

Corrected

Line 298: Please mention the Caption after the figure (for all figures)

This is according to the submission guide.

Line 380: Please delete the word ‘Footnote’ from all the tables. Please write the footnote below the Tables.

Corrected in all the tables.

Line 384: *P < 0.05 or *P ≤0.05?

Corrected (line 444)

Line 478: Please revise (Table 2)

Without any further explanation why or what in the Table needed revision, we have done some minor edits but left it essentially the same.

Please mention a line of future implications of the output in the conclusion.

Response: Have added: ‘Future gene editing or genetic engineering related to NIK3/PTI in farmer-preferred or commercially grown cassava cultivars could improve tolerance/basal immunity to CMD and consequently starch yield (line 737).

Please cross-check the language, grammar, and punctuation.

Response: Have done so.

Please cross-check the reference pattern as per the journal standard.

Response: Done.

Line 688: References…please check the section numbering per journal pattern.

Response: Corrected section numbering.

The number of references (86) may be reduced.

Response: Have kept as is as many different topics are covered in this MS.

Please use high-resolution images in the Figures/graphs.

Response: The figure resolution is 600dpi generally considered as an acceptable figure resolution for publications, and the text on figures has been increased for better readability.

The manuscript is well-prepared per the journal standard and may be considered with minor revision. Good luck with the revision.

Response: Thank you for your invaluable input.

Reviewer #2: The author discussed the physiological changes in plants when inoculated with South African cassava mosaic virus (SACMV) using a susceptible and non-susceptible (tolerant) cassava variety.

In particular, this paper discussed the plant immune response that involves a mutual potentiation of cell membrane receptor-associated pattern-triggered immunity (PTI) and nucleotide leucine-rich repeat (NLR) effector-associated immunity (ETI) in plant immune responses when inoculated with SACMV.

It is interesting that the amplification process of SACMV was investigated in detail at each day after inoculation, physiological investigations such as quantification of ROS were also conducted, and the discussion was performed from various perspectives.

Response/clarification: Investigations were done at three time points (12, 32 and 67 days post the initial SA CMV agro-inoculation.

However, although the gene at the CMD2 locus, which functions most effectively in SLCMV resistance, has been identified in recent years, there is a lack of discussion on the relationship between mutations on CMD2 locus and the level of SLCMV symptoms.

The author needs to discuss two peroxidases near the CMD2 locus in the discussion section. Do these two types of gene changes occur during SLCMV infection?

Response: We have discussed CMD1,2,3 lines 688 - line 701.

We added Kungu’s et al., 2024 [93] report on ‘Reduction of cassava mosaic geminiviruses from infected stem cuttings using salicylic acid, hydrogen peroxide and hot water treatment’. We also referenced Wang et al 2020 [94] report on ‘WRKYs binding directly to the W-box-containing of RBOHD promoting ROS accumulation in plant-bacterium resistance’. Also, Fig. 5 was updated by adding WRKYs effect on ROS production.

Also, are there any special SNPs in the genes near the CMD2 locus? I think a partial revision of the Discussion section is necessary.

Response: In this current study, no CMD1-2 loci sequencing and mapping was conducted in T200 or TME3. As a result, we cannot discuss SNPs in the genes located in CMD2 region.

According to the literature, a nonsynonymous mutation (SNP) in DNA polymerase

beta subunit 1 (MePOLD1) within the CMD2 region is linked to CMD2-type resistance (doi.org/10.1038/s41467-022-31414-0). This is discussed adequately in the introduction section. However, many HOST genes besides DNA polymerase contribute to the global response to cassava geminiviruses in both susceptible, tolerant and resistant germplasm. There are also quantitative responses to CMD which MePOLD does not explain.

Regarding the quantitative of SMART sequencing,

The number of HiFi Reads in Table 2 varies greatly among samples. Please explain whether this data reflects the amount of transcription and what statistical processing was performed.

Response: Table 3 (then Table 2) data was generated from genome DNA amplicons amplified using targeted primers. It is not a gene expression profiling data of the cell-surface LRR receptor kinases (LRR-RKs), cell-surface LRR receptor protein (LRR-RP) and intracellular nucleotide-binding receptors (NLRS). The variation in HiFi reads indicates different genome architecture of the cassava germlines, and in essence confirms high gene variation of LRR-RP, LRR-RK and NLRs in cassava.

The high number of unknown (unmapped) reads when mapped on the reference genome is a good indication that (1) sample preparation and (2) sequence runs were successful.

Please increase the resolution of the figures. The text in Figures 2b, 2c, and 2d, and Figs. 3a, 3b, and 3c are difficult to see.

Response: Figures resolution is 600 dpi.

Text (font size) in Figures increased.

Please show the information of original or raw data of SMART sequencing obtained from Gene Expression Omnibus etc.

Response: SMRT raw data and has been uploaded and available under the BioProject number PRJNA1190476 (which is included in the manuscript).

---

## [Decision Letter · Decision Letter 1]

16 Jan 2025

Comparative analysis of pattern-triggered and effector-triggered immunity gene expression in susceptible and tolerant cassava genotypes following begomovirus infection

PONE-D-24-29673R1

Dear Dr. Rey,

We’re pleased to inform you that your manuscript has been judged scientifically suitable for publication and will be formally accepted for publication once it meets all outstanding technical requirements.

Kind regards,

S.V.Ramesh, PhD

Academic Editor

PLOS ONE

Additional Editor Comments (optional):

Reviewers' comments:

Reviewer's Responses to Questions

**Comments to the Author**

1. If the authors have adequately addressed your comments raised in a previous round of review and you feel that this manuscript is now acceptable for publication, you may indicate that here to bypass the “Comments to the Author” section, enter your conflict of interest statement in the “Confidential to Editor” section, and submit your "Accept" recommendation.

Reviewer #2: All comments have been addressed

2. Is the manuscript technically sound, and do the data support the conclusions?

Reviewer #2: Yes

3. Has the statistical analysis been performed appropriately and rigorously? 

Reviewer #2: I Don't Know

4. Have the authors made all data underlying the findings in their manuscript fully available?

Reviewer #2: Yes

5. Is the manuscript presented in an intelligible fashion and written in standard English?

Reviewer #2: Yes

6. Review Comments to the Author

Reviewer #2: Your responses to the comments are detailed, accurate, and well-writing, addressing the points raised comprehensively. I appreciate the clarity and effort you have put into them. Therefore, I will accept it.

7. PLOS authors have the option to publish the peer review history of their article (what does this mean? ). If published, this will include your full peer review and any attached files.

**Do you want your identity to be public for this peer review?** For information about this choice, including consent withdrawal, please see our Privacy Policy .

Reviewer #2: No

---

## [Editor Report · Acceptance letter]

PONE-D-24-29673R1

PLOS ONE

Dear Dr. Rey,

I'm pleased to inform you that your manuscript has been deemed suitable for publication in PLOS ONE. Congratulations! Your manuscript is now being handed over to our production team.

Kind regards,

on behalf of

Dr. Shunmugiah Veluchamy Ramesh

Academic Editor

PLOS ONE